# Energy-Based Attribute Models for Controllable Review Generation with Frozen LLMs

## Abstract

Large language models (LLMs) generate fluent text but, as monolithic systems, lack native mechanisms for fine-grained attribute control (e.g., clamping a single attribute such as brand preference while holding everything else fixed). We propose an architectural separation: a frozen LLM (Qwen2.5-0.5B-Instruct) provides linguistic competence, while an external energy-based attribute model (Deep Boltzmann Machine, DBM) captures the domain's attribute co-occurrence structure. The hidden vector of DBM is projected into soft prompts by a lightweight adapter and rendered as text by a frozen LLM. We use this controlled rendering architecture to generate review text consistent with a desired attribute configuration. This split yields a falsifiable prediction: a direct-MLP baseline (the proposed pipeline without the DBM) should match or marginally outperform the proposed model on surface metrics (CE, SBERT), while structure-level capabilities, such as energy-based coherence scoring, predictively valid clamping, and independent two-axis attribute control, should be exclusive to the energy-based pipeline. Both halves resolve as predicted on two Amazon review domains (Smartphones, Beauty) under a 3-seed protocol. The proposed model statistically dominates LoRA, full fine-tuning, CTRL, PPLM, and one-shot ICL on surface metrics and matches or is marginally outperformed by direct-MLP; on structure-level metrics it beats direct-MLP on every cell: its energy response tracks each domain's empirical coupling structure (association +0.94/+0.80 over all 21 attribute pairs), coupling-aware combined clamps cut Wasserstein distance to a model-consistent low-rating reference by 23% on Beauty, and two-axis clamps yield 1.6–3.4 times larger content shifts with sentiment held fixed. An RBM ablation isolates the role of depth: a single hidden layer matches the DBM on form, coupling recovery, and sentiment-axis clamps, but only hierarchical latents preserve independent two-axis control. A generously tuned activation-steering baseline (CAA) matches the raw sentiment shift but fails distributional match to natural reviews, composes destructively, and entangles the axes. Clamping is model-internal control—consistency with the learnt joint distribution—and implies no real-world causal claim; within that scope, architectural separation offers a practical path toward controllable, interpretable review generation with frozen LLMs.

## 1 Introduction

Large Language Models (LLMs) have emerged as remarkably capable language generators. Yet, as monolithic systems, they do not natively expose mechanisms for fine-grained clamping of individual latent concepts (e.g., what the model would generate if a single attribute such as brand preference were clamped to a different value while holding everything else fixed). All knowledge remains entangled within a single set of parameters (Cloud et al., 2025). Existing conditional generation methods, such as control codes (Keskar et al., 2019) and gradient-based steering (Dathathri et al., 2020), rely on correlational signals rather than structured, inspectable representations of the underlying domain.

The broader question of whether LLMs can genuinely model the world has gained renewed attention. Leading researchers have argued that autoregressive LLMs are structurally limited in their ability to model causality and world structure, and that capable AI systems require explicit world models that predict in abstract

representation space rather than token space (LeCun, 2022; Ha & Schmidhuber, 2018). Our architecture takes inspiration from this debate: the component that generates language need not be the component that represents domain structure — *the mouth is not the brain.*

We therefore propose a modular architecture that explicitly separates the two: (i) a Deep Boltzmann Machine (DBM; Salakhutdinov & Hinton, 2009) serves as an energy-based attribute model that captures the domain's latent co-occurrence structure; (ii) a lightweight MLP adapter projects the DBM's mean-field beliefs into soft prompt embeddings; (iii) a frozen modern LLM provides linguistic competence, receiving only the minimal textual prompt for the format. By externalizing domain knowledge into a dedicated, inspectable attribute model, this separation naturally yields two critical properties absent from standard monolithic adaptation: **interpretability** (the system's beliefs can be directly examined via the energy-based model) and **controllability** (specific latent factors can be selectively clamped, and the clamp propagates through the learnt joint distribution).

For domains that exhibit structured latent regularities, our position is that **structured attribute modeling and language generation should be architecturally separated**: an inspectable energy-based attribute model should condition a frozen language model through a lightweight adapter. As an organizing analogy we borrow *form/meaning distinction* (Bender & Koller, 2020): **form** (linguistic surface) is the competence of the LLM, while the structured attribute configuration on which clamps act plays the role of **meaning** and is captured by the external attribute model.

This architectural split yields a clear, testable prediction. A strong form-only baseline that optimizes linguistic quality end-to-end with direct access to visible features (B1: direct-MLP adapter, see §4) is expected, by construction, to match or marginally outperform the proposed model on surface-level (form) metrics such as cross-entropy and SBERT similarity. In contrast, structure-level capabilities (energy-based coherence scoring, predictively valid attribute clamps, and independent two-axis clamp-based control) should be exclusive to the DBM-based pipeline.

We test and confirm this prediction using a 2.71M-parameter DBM (§4) with a frozen open-weight LLM (Qwen2.5-0.5B-Instruct). Four lines of evidence support the framework. **1. Form:** under the 3-seed protocol the proposed model statistically dominates LoRA, full fine-tuning, CTRL (Keskar et al., 2019), PPLM (Dathathri et al., 2020), and one-shot ICL, and matches or is marginally outperformed by direct-MLP, as predicted (§5). **2. Coupling discovery:** the DBM's energy response tracks each domain's empirical coupling structure across all 21 attribute pairs (§6). **3. Two-axis control:** cross-domain invariant couplings (Rating–Service) give a sentiment handle and domain-specific couplings (Brand–Topic specialty) a content handle; coupling-aware combined clamps sharpen the match to the model-consistent low-rating reference, while a generously tuned activation-steering baseline (CAA; Rimsky et al. 2024) fails distributional match, composes destructively, and entangles the two axes (§7–§7.4). **4. Ablations:** removing the energy model entirely (direct-MLP, which can be re-conditioned but not clamped) and removing only its depth (RBM) stratify the capabilities: an energy model is required for coherence scoring and clamp-based generation, a latent joint-density model of any depth suffices for form and sentiment-axis clamps, and hierarchical latents are required for content-axis strength and two-axis orthogonality (§8).

## 2 Related Work

Energy-based models (EBMs; LeCun et al., 2006) score configurations with an energy function in which low energy corresponds to plausible configurations, and the world-model literature has advocated them as a natural substrate for representing domain structure for environment dynamics (Schmidhuber, 2015; Ha & Schmidhuber, 2018) or for prediction in abstract representation space (LeCun, 2022; Assran et al., 2023; Bardes et al., 2023). We borrow only the energy-based formulation from this lineage and apply it to **static co-occurrence market structure**: which attribute configurations are coherent within a given market, with no temporal dynamics. Among EBMs, Boltzmann machines (BMs; Ackley et al., 1985) learn distributions over binary variables via an energy function. Restricted Boltzmann Machines (RBMs; Smolensky, 1986; Hinton, 2012) remove intra-layer connections to enable efficient mean-field inference and Contrastive Divergence (CD) training. Deep Boltzmann Machines (DBMs; Salakhutdinov & Hinton, 2009) further stack multiple RBMs with undirected connections, supporting bidirectional inference. While DBMs have been

applied to multimodal learning (Ngiam et al., 2011; Srivastava & Salakhutdinov, 2012), to our knowledge, they have not been used as an explicit, inspectable attribute model that conditions a frozen LLM. Compared with variational autoencoders (VAEs; Kingma & Welling, 2014), which learn continuous latents via a reconstruction-plus-KL objective, DBMs offer a distinct advantage: they provide a natural energy function that directly quantifies the *coherence* of arbitrary input configurations. This combination of energy-based evaluation, generative capability, and structured latent representations makes DBMs particularly well-suited as inspectable structured attribute models.

A complementary line uses the LLM itself as the model of the domain: generative agents and social simulation (Park et al., 2023; Yan et al., 2024; Bougie & Watanabe, 2025) elicit coherent populations and environments from an LLM's implicit knowledge, and LLM-JEPA (Huang et al., 2025) adds a JEPA-style auxiliary loss during fine-tuning as an implicit structural regularizer. In these systems the domain structure lives inside the LLM, where it is neither inspectable nor clampable; we instead train an explicit external model of attribute structure and condition a frozen LLM on it through learned soft prompts.

For controllable generation and PEFT, classic methods such as CTRL (Keskar et al., 2019) prepend control codes, while PPLM (Dathathri et al., 2020) steers a frozen LM via attribute-classifier gradients. Both operate on individual attributes without joint modeling of correlational structure. Among parameter-efficient fine-tuning methods, LoRA (Hu et al., 2022) has become the standard modern baseline. In addition, soft prompt tuning (Li & Liang, 2021) learns prompts end-to-end from text; we differ in deriving prompts from an external attribute model rather than from text alone. A complementary line controls frozen LLMs by editing their internal representations. Activation steering methods, such as activation addition (Turner et al., 2023), contrastive activation addition (Rimsky et al., 2024), inference-time intervention (Li et al., 2023), and function/in-context vectors (Todd et al., 2024; Hendel et al., 2023; Liu et al.), extract a direction in the LLM's activation space (typically a contrastive difference-of-means or linear probe) and add it back at inference time; representation engineering (Zou et al., 2023) systematizes the recipe on the longstanding linear-direction view (Kim et al., 2018; Park et al., 2024). Sparse-autoencoder steering clamps interpretable SAE features (Templeton et al., 2026; Gao et al., 2025; Chalnev et al., 2024), and DExperts (Liu et al., 2021) reshapes logits with auxiliary LMs at decoding time. In all cases the steering signal is a single correlational direction (or feature) per attribute applied inside the frozen LLM, with no separate jointly-trained model of how attributes co-occur. Our controllable object is instead an external energy-based attribute model whose binary latents encode attribute co-occurrence; clamping visible units and re-running mean-field inference propagates the clamp through an inspectable joint distribution while the LLM's weights and activations remain untouched.

Finally, regarding consumer review generation, early work used RNNs with attention mechanisms (Dong et al., 2017; Ni & McAuley, 2018). More recent approaches construct prompts from user behavioral history and fine-tune LLMs (Peng et al., 2024). A central challenge is *consumer heterogeneity* along latent difference such as price sensitivity and brand loyalty (Smith, 1956; DeSarbo et al., 1997; Wedel & Kamakura, 2000) which cannot be observed from the surface behavior (Tseng et al., 2024; Zollo et al., 2025). LLM-based persona generation often reproduces systematic biases relative to true population distributions (Li et al., 2025). Our approach instead learns latent representations of heterogeneity as a market structure directly from behavioral data through an energy-based attribute model.

## 3 Architecture

The proposed architecture consists of three components (Fig. 1): (i) a DBM that serves as an energy-based attribute model, learning the latent co-occurrence structure of a domain from binary visible features, (ii) a lightweight MLP adapter that projects the DBM's mean-field beliefs into soft-prompt embeddings, and (iii) a frozen LLM that renders those beliefs into fluent natural language.

The DBM comprises one visible layer $\mathbf{v}$ with $J_v$ binary units and $L$ hidden layers $\mathbf{h}^{(l)}$ with $J_l$ units each $(l = 1, \ldots, L)$. The energy function is defined as

$$E(\mathbf{v}, \mathbf{h}^{(1)}, \ldots, \mathbf{h}^{(L)}) = -\sum_{l=1}^{L} \mathbf{h}^{(l-1)\top} W^{(l)} \mathbf{h}^{(l)} - \mathbf{b}^\top \mathbf{v} - \sum_{l=1}^{L} \mathbf{c}^{(l)\top} \mathbf{h}^{(l)}, \tag{1}$$

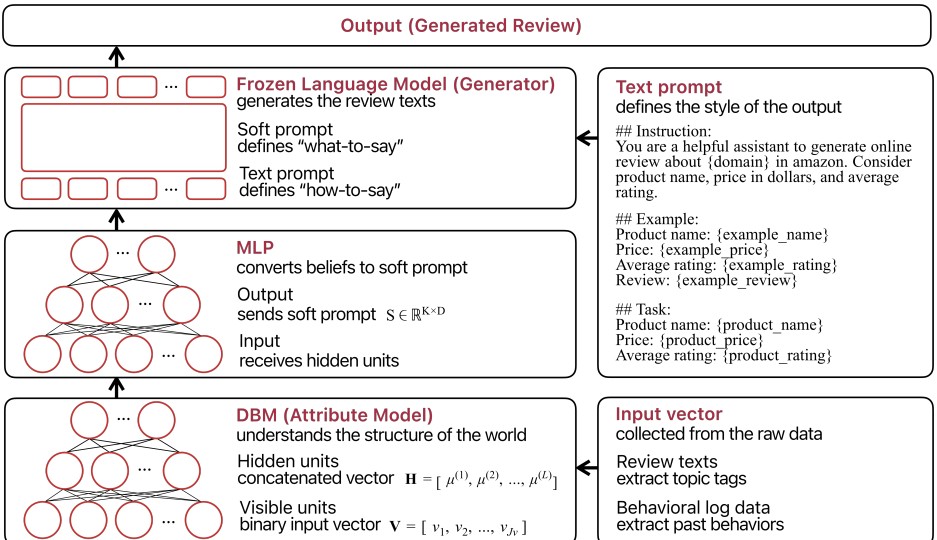

Figure 1: Overview of the three-component architecture. The DBM is trained unsupervised on binary visible features extracted from reviews. The adapter and LLM never observe the raw tabular features directly. During the final training phase, only the adapter is trained while both the DBM and LLM remain frozen.

where $\mathbf{h}^{(0)} \equiv \mathbf{v}$. Posterior inference uses a fully factorized mean-field approximation $q(\mathbf{h}) = \prod_{l,i} \mathrm{Bern}\big(h_i^{(l)}; \mu_i^{(l)}\big)$; thanks to the bipartite structure between consecutive layers, its means $\mu^{(l)}$ satisfy the fixed-point iteration

$$\mu^{(l)} \leftarrow \sigma\Big(W^{(l)\top}\mu^{(l-1)} + W^{(l+1)\top}\mu^{(l+1)} + \mathbf{c}^{(l)}\Big), \qquad (2)$$

where $\sigma$ is the element-wise logistic sigmoid, with boundary conditions $\mu^{(0)} \equiv \mathbf{v}$ and $W^{(L+1)}\mu^{(L+1)} \equiv \mathbf{0}$. At inference time, we use the deterministic mean-field activations rather than stochastic samples for adapter conditioning. The concatenated vector $\mathbf{H} = [\mu^{(1)}; \dots; \mu^{(L)}] \in \mathbb{R}^{J_1 + \dots + J_L}$ serves as the abstract *belief* representation that is passed to the adapter. The mean-field energy score $\tilde{F}(\mathbf{v})$, used as a coherence score, is defined in App. A.

A two-layer MLP adapter then maps the belief vector $\mathbf{H}$ to a sequence of $K$ soft-prompt embeddings $\mathbf{S} \in \mathbb{R}^{K \times D}$, prepended to the LLM's input sequence. Only the adapter parameters are updated during the final training phase, using the AdaMax optimizer (Kingma & Ba, 2015) and cross-entropy loss with respect to ground-truth reviews.

Finally, we employ Qwen2.5-0.5B-Instruct as the frozen generator, with all parameters kept frozen throughout training and inference. The LLM receives only a minimal textual product context (product name, price, and average rating) shared with every baseline. All additional structured domain knowledge is supplied exclusively through the DBM-derived soft prompt. The exact prompt template is provided in App. B.

## 4   Experimental Design

**Dataset.** We conduct experiments on Amazon product reviews (McAuley et al., 2015; He & McAuley, 2016) from two distinct domains: Smartphones (electronics category with oligopolistic brand structure) and Beauty products (long-tail category with brand–topic specialization). We use English reviews from verified purchases with lengths between 100 and 1024 characters (detected via FastText (Bojanowski et al., 2017; Joulin et al., 2017)). The dataset is split into train/val/test = 52,952/1,024/1,024 per domain. Each consumer profile is represented as a binary one-hot vector of dimension $J_v = 160$ (Smartphones) or $J_v = 170$ (Beauty), encoding brand, price tier, rating, and additional contextual signals (e.g., repeat buyer, accessory purchases) derived from the user's prior review history.

**Inference.** Given visible **v**, we run mean-field iteration to convergence, project beliefs through the adapter to soft prompts, prepend to a one-shot text prompt, and generate text autoregressively with the frozen LLM (temperature 0.7, max_new_tokens = 100). Detailed hyperparameters for the three training phases (LPT, JFT, and adapter training) are provided in App. E; preprocessing and feature construction are detailed in App. D.

**Baselines.** Seven baselines, all built on Qwen2.5-0.5B-Instruct as the base language model: **B0** Frozen In-Context Learning (ICL) with a one-shot example; **B1** Direct-MLP adapter: an ablation of the proposed model that keeps the conditioning channel intact (i.e., the same visible vector, a nonlinear MLP mapping to soft prompts, a frozen generator, and an identical text prompt) but removes the generatively trained joint-density model (no energy function, no clampable belief space), serving as the leakage-controlled form-side comparator; **RBM** Depth ablation of the proposed model: a single-hidden-layer Boltzmann machine with total hidden dimensionality matched to the DBM's (3584 units), so the belief vector **H** and the adapter are identical and only latent depth is ablated ($\approx$0.58M vs. 2.71M energy-model parameters; training protocol, data, and seeds identical; App. F); **B2** Full fine-tuning of all LLM parameters on the in-domain training corpus; **B3** LoRA (rank 8; Hu et al., 2022) applied to the LLM; **CTRL** (Keskar et al., 2019) controlling codes prepended at decoding time; **PPLM** (Dathathri et al., 2020) attribute-classifier-guided steering. The proposed model, B0, B1, the RBM ablation, CTRL, and PPLM keep the generator frozen. B1's MLP is trained end-to-end on the LLM's CE loss with direct access to visible features, giving it a structural advantage on form metrics that we analyze in §8.

**Metrics.** We evaluate form-level quality using test-set cross-entropy (CE; lower is better) and SBERT cosine similarity (CosSim; higher is better) on $n = 1{,}024$ held-out reviews. Statistical significance is assessed via paired $t$-tests on item-level differences (and Wilcoxon signed-rank tests where noted); see the multi-seed protocol below. Sentiment is measured with a RoBERTa sentiment classifier (`cardiffnlp/twitter-roberta-base-sentiment-latest`; scored as $P(\text{positive}) - P(\text{negative}) \in [-1, 1]$).

**Multi-seed protocol.** All quantitative results use a 3-seed protocol that separates model variance from data variance. (i) **Fixed test split**: The data split seed is fixed at 0, so the same 1,024 test reviews (and matched clamped samples) are used across all seeds. (ii) **Training/generation seed**: For trainable models (Proposed, RBM, B1, B3), we vary the random seed $\in \{0, 1, 2\}$, affecting Boltzmann-machine training (PCD sampling and initialization), DataLoader shuffling, and decoding. For non-trainable baselines, only the generation seed varies. (iii) **Statistical inference**: We report 3-seed mean $\pm$ std. Because the test split is fixed, we average each item's per-seed paired differences and run paired tests on the resulting $n$ *item-level* differences (paired $t$; Wilcoxon signed-rank as a non-parametric cross-check), making items rather than seed-replicates the unit of inference. Pooling all $3 \times n$ per-sample differences instead yields the same significance pattern on every reported claim.

**Conditioning and feature provenance.** Visible features consist of *history-derived* signals (e.g., purchase frequency, repeat-buyer status) and *auto-extracted topical* features from the target review. This setup is best interpreted as *controlled rendering of text given desired attributes* rather than pure next-review prediction. All baselines receive the same per-sample attribute information as the proposed model, but through different channels: text fields appended to the prompt (B0/B2/B3/CTRL/PPLM) and the visible vector passed through the adapter (Proposed/RBM/B1). These channels differ in conditioning bandwidth and inductive bias: a serialized text field is bounded by the prompt interface, while the adapter path delivers a learned continuous projection of the same features. B1—which shares the visible-vector channel, the adapter architecture, and the frozen generator—is therefore the leakage-controlled, fair form-side comparator, and we treat it as such at every comparison site; the text-channel baselines contextualize against standard practice rather than isolate the DBM's contribution.

## 5 Experiment I: Generation Quality

We first verify that the proposed pipeline does not sacrifice form-side generation quality relative to standard adaptation methods. Table 1 reports the main comparison. Across 3 seeds the proposed model dominates B0/B2/B3/CTRL/PPLM on every cell in both domains ($p < 10^{-10}$ on every cell except B2 Smartphone CE, $p{=}5.1{\times}10^{-6}$). As expected, B1 (direct MLP, no DBM) wins one form metric per domain (Smartphone

CosSim and Beauty CE) by margins within seed-to-seed std. The RBM wins no cell against the proposed model: the DBM is significantly better on Smartphone CE ($p < 10^{-10}$) and Beauty CosSim ($p$=0.014), with the remaining two cells statistically tied—latent depth contributes modestly but consistently on the form side. The dominance over Full FT and LoRA in particular establishes that DBM-mediated conditioning does not compromise generation quality.

Table 1: Generation quality on Smartphone and Beauty review generation. CE and CosSim on $n$=1024 test reviews per seed (PPLM rows drop 1–4% of failed generations); 3-seed protocol per §4, mean $\pm$ std; $p$-values are item-level paired $t$-tests against the proposed model (per-item differences averaged across the 3 seeds; §4). Since PPLM's dropped generations break the item-level pairing for CosSim, CosSim $p$ is the per-seed maximum (every seed $< 10^{-10}$).

| Model | Smartphone | | | | Beauty | | | |
|---|---|---|---|---|---|---|---|---|
| | CE | $p$ | CosSim | $p$ | CE | $p$ | CosSim | $p$ |
| **Proposed** (DBM + adapter) | **3.135 ± 0.005** | — | 0.534 ± 0.007 | — | 3.181 ± 0.006 | — | **0.455 ± 0.005** | — |
| RBM ablation | 3.169 ± 0.002 | $< 10^{-10}$ | 0.538 ± 0.004 | 0.051 | 3.181 ± 0.009 | 0.772 | 0.449 ± 0.006 | 0.014 |
| B1: Direct MLP (no DBM) | 3.137 ± 0.003 | 0.383 | **0.541 ± 0.002** | $1.3 \times 10^{-3}$ | **3.175 ± 0.003** | $2.9 \times 10^{-3}$ | 0.453 ± 0.003 | 0.354 |
| B2: Full fine-tuning | 3.161 ± 0.003 | $5.1 \times 10^{-6}$ | 0.438 ± 0.003 | $< 10^{-10}$ | 3.261 ± 0.009 | $< 10^{-10}$ | 0.407 ± 0.007 | $< 10^{-10}$ |
| B3: LoRA (Hu et al., 2022) | 3.197 ± 0.001 | $< 10^{-10}$ | 0.440 ± 0.002 | $< 10^{-10}$ | 3.240 ± 0.001 | $< 10^{-10}$ | 0.405 ± 0.002 | $< 10^{-10}$ |
| CTRL (Keskar et al., 2019) | 3.473 ± 0.000 | $< 10^{-10}$ | 0.371 ± 0.002 | $< 10^{-10}$ | 3.418 ± 0.000 | $< 10^{-10}$ | 0.364 ± 0.001 | $< 10^{-10}$ |
| PPLM (Dathathri et al., 2020) | 3.583 ± 0.000 | $< 10^{-10}$ | 0.275 ± 0.002 | $< 10^{-10}$ | 3.583 ± 0.000 | $< 10^{-10}$ | 0.233 ± 0.006 | $< 10^{-10}$ |
| B0: One-shot ICL | 3.674 ± 0.005 | $< 10^{-10}$ | 0.421 ± 0.002 | $< 10^{-10}$ | 3.619 ± 0.004 | $< 10^{-10}$ | 0.401 ± 0.000 | $< 10^{-10}$ |

# 6 Experiment II: Coupling Discovery as Attribute-Model Validation

We second conduct Experiment II for attribute-model validation. In Phase 1 (discovery), for all $\binom{7}{2}$=21 cross-prefix visible-attribute pairs in each domain, we rank empirical Pearson correlations and select the strongest cross-prefix couplings. In Phase 2 (validation), we test whether the DBM's energy landscape reflects each coupling: for every cross-prefix pair we compute a base-rate-independent *interaction contrast* of the mean-field energy score and relate it to the empirical correlation across all 21 pairs.

**Phase 1: empirical coupling discovery.** Table 2 shows that the strongest cross-prefix couplings differ markedly between domains (full rankings in App. L). Smartphone exhibits an oligopolistic Brand–Price/Purchase coupling, consistent with a market where a few brands occupy distinct price tiers. Beauty shows weaker price coupling but distinctive brand–category specialization (e.g., Sally Hansen–Color, Bath & Body Works–Fragrance), consistent with a more fragmented market where brand identity is anchored in product specialty rather than price tier. Across both domains, Rating–Topic(SERVICE) is consistently coupled, suggesting a cross-domain invariant rooted in consumer psychology.

Table 2: Top empirical couplings (max $|r|$ across all unit pairs within prefix combinations). Brand–Price coupling differs by 3.6× between domains; Rating–Topic coupling is invariant.

| Coupling | Type | Smartphone $|r|$ | Beauty $|r|$ |
|---|---|---|---|
| PriceRange × Purchase | oligopoly (domain-specific) | 0.908 | 0.289 |
| Brand × Purchase | oligopoly (domain-specific) | 0.636 | 0.027 |
| Brand × Price | oligopoly (domain-specific) | 0.277 | 0.076 |
| Rating × Topic(SERVICE) | cross-domain invariant | 0.254 | 0.212 |
| Brand × Topic specialty [†] | long-tail (domain-specific) | 0.249 | 0.161 |

[†] Apple–Damage (Smartphone), Sally Hansen–Color (Beauty).

**Phase 2: energy response tracks empirical coupling.** For each of the 21 cross-prefix pairs $(a, b)$ (representative unit pair per prefix block, chosen by max $|r|$), we compute the signed interaction contrast

$$d(a,b) \;=\; \big[\tilde{F}(a{=}1, b{=}0) + \tilde{F}(a{=}0, b{=}1)\big] \;-\; \big[\tilde{F}(a{=}1, b{=}1) + \tilde{F}(a{=}0, b{=}0)\big], \tag{3}$$

averaged over data-derived base configurations and normalized by the per-seed energy std; $d(a,b) > 0$ iff the aligned configurations $(1,1)/(0,0)$ receive lower energy, so $d$ should mirror the empirical coupling $r$, and

unlike per-direction percentage flips it is not confounded by marginal base rates (App. L). Table 3 reports the association across all 21 pairs: the DBM's energy response tracks the empirical coupling structure (Pearson $+0.94/+0.80$, permutation $p=2\times10^{-4}$), with the RBM ablation—scored with its exact free energy (App. F)— close behind: coupling recovery does not require depth. The conditional lift $P(b\,|\,a{=}1) - P(b)$ matches the ranking near-tautologically: coupling *discovery* needs no energy model at all. What the energy model adds is generative—a coherence score over arbitrary configurations (Table 13) and clamp-and-re-equilibrate inference (§7)—neither available to a correlation table.

Table 3: Energy response vs. empirical coupling across all 21 pairs. Correlations between the interaction contrast $d(a, b)$ and the empirical coupling $r$; Pearson is the 3-seed mean $\pm$ std of per-seed correlations, Spearman is computed once on the seed-averaged response, and perm. $p$ is a two-sided permutation test (5,000 label shuffles). The correlation-lift baseline is a deterministic re-description of the data.

| Domain | Scorer | Pearson | Spearman | perm. $p$ |
|---|---|---|---|---|
| Smartphone | DBM mean-field energy | $+0.941 \pm 0.013$ | $+0.921$ | $2 \times 10^{-4}$ |
| | RBM exact free energy | $+0.912 \pm 0.002$ | $+0.929$ | $2 \times 10^{-4}$ |
| | Correlation lift $P(b\,|\,a{=}1) - P(b)$ | $+0.917$ | $+0.977$ | $2 \times 10^{-4}$ |
| Beauty | DBM mean-field energy | $+0.802 \pm 0.020$ | $+0.832$ | $2 \times 10^{-4}$ |
| | RBM exact free energy | $+0.724 \pm 0.079$ | $+0.501$ | $1.4 \times 10^{-3}$ |
| | Correlation lift $P(b\,|\,a{=}1) - P(b)$ | $+0.950$ | $+0.914$ | $2 \times 10^{-4}$ |

Residual leakage is reported rather than claimed away: the association is strong but imperfect, and individual weakly-coupled pairs can still draw a non-trivial energy response—most visibly Beauty's catch-all Brand(OTHERS) × Purchase($\geq$\$100) pair ($r{=}0.03$, normalized $d{=}{+}0.10$, above the median response of appreciably coupled pairs) and, on Smartphone, Price(ENTRY) × Rating(5) ($r{=}{-}0.05$, $|d|$ comparable to the weakest coupled pairs). These deviations are what the permutation test tolerates; we flag them as the current fidelity limit of the learnt structure.

**Three-tier latent structure.** Combining Phases 1 and 2 yields a three-tier organization of the learnt belief space: **(i) cross-domain invariant** (Rating–Service in both domains, $r{\sim}0.21$–$0.25$), reflecting consumer-psychology regularities; **(ii) domain-specific** (Smartphone Brand–Price oligopoly, Beauty Brand–Topic specialty), reflecting market structure; **(iii) weakly-coupled** (empirical $r \approx 0$), where the energy response is correspondingly small—a fidelity quantified by the continuous association of Table 3 rather than by a binary near-null claim, with the residual deviations reported above. The DBM does not impose rigid priors; it learns whichever coupling structure each domain exhibits. App. M shows the full Pearson correlation matrices of visible variables under both the empirical distribution and DBM samples, confirming that the DBM tracks each domain's second-order co-occurrence structure rather than imposing a shared prior.

**Energy ranks natural over swap-perturbed configurations.** Beyond per-pair flips, the DBM's mean-field energy score globally ranks naturally observed configurations as more plausible than swap-perturbed ones (LeCun et al., 2006). For each of the six single-attribute clamps used in §7, we compare $\tilde{F}(v_{\text{original}})$ to $\tilde{F}(v_{\text{clamped}})$ where one attribute is swapped to a less-coupled value (Table 13, App. L). $\Delta\bar{F}$ is positive in all six conditions and the natural configuration receives the lower energy on 84–99% of paired samples; the gap scales with empirical coupling strength (largest for Smartphone Price Premium→Entry, $+11.9$; smallest for Smartphone Brand swap, $+3.9$).

## 7 Experiment III: Coupling-Aware Targeted Clamp

The latent structure uncovered in Experiment II provides two distinct clamp handles: *sentiment-axis* clamps exploit cross-domain invariant couplings (e.g., Rating–Service), while *content-axis* clamps exploit domain-specific couplings (e.g., Brand–Topic specialty). We therefore verify their independence, using several baselines: B1, the RBM, and a generously tuned activation-steering baseline (CAA; Rimsky et al. 2024) (§7.4).

All clamp effects reported below are additionally validated with a review-domain star classifier and a zero-shot topic classifier, which reproduce the same orderings (App. I)[1].

## 7.1 Single-attribute selectivity

We apply three single-attribute clamps per domain (Rating $5 \rightarrow 1$; Price Premium/Luxury $\rightarrow$ Entry/Budget; Brand Apple$\rightarrow$Samsung, MAC$\rightarrow$L'Oréal) under the 3-seed evaluation, measuring sentiment with RoBERTa.

Table 4 shows the results. The Rating clamp produces a strong sentiment shift in both domains for all three models, while most Price/Brand cells remain null. The one Price/Brand cell where the DBM resolves a signal is Beauty Price, consistent with the empirical Brand–Price coupling ($r=0.076$) being weak but non-zero; the RBM and B1 both miss it. Selectivity therefore reflects fidelity to the data: where the empirical coupling is appreciable the clamp leaves a detectable signature, where it is absent none is induced, and sensitivity to the weakest couplings is the first place depth pays. The one deviation is B1's marginal Smartphone Price drift, which tracks no appreciable empirical coupling.

Table 4: Single-attribute clamp effects on sentiment under 3-seed evaluation. The paired-$t$ test is computed against zero on item-level differences (each item's per-seed differences averaged across the 3 seeds). †: paired-$t$ $p < 10^{-3}$; *: $p < 0.05$; ∘: $p < 0.1$; ns: $p \geq 0.1$ with Wilcoxon signed-rank test.

| Domain | Clamp | Model | $n_{\text{per-seed}}$ | $\Delta$ RoBERTa | paired-$t$ $p$ | Wilcoxon $p$ |
|---|---|---|---|---|---|---|
| Smartphone | Rating $5\rightarrow1$ | Proposed | 200 | $-1.388\pm0.045$† | $< 10^{-10}$ | $< 10^{-10}$ |
| | | RBM | 200 | $-1.209\pm0.096$† | $< 10^{-10}$ | $< 10^{-10}$ |
| | | B1 | 200 | $-1.279\pm0.069$† | $< 10^{-10}$ | $< 10^{-10}$ |
| | Price Premium$\rightarrow$Entry | Proposed | 100 | $+0.038\pm0.028^{ns}$ | 0.250 | 0.415 |
| | | RBM | 100 | $+0.001\pm0.055^{ns}$ | 0.979 | 0.921 |
| | | B1 | 100 | $+0.060\pm0.056^{\circ}$ | 0.061 | 0.013 |
| | Brand Apple$\rightarrow$Samsung | Proposed | 100 | $+0.058\pm0.029^{\circ}$ | 0.056 | 0.183 |
| | | RBM | 100 | $+0.051\pm0.013^{\circ}$ | 0.094 | 0.131 |
| | | B1 | 100 | $+0.001\pm0.016^{ns}$ | 0.964 | 0.989 |
| Beauty | Rating $5\rightarrow1$ | Proposed | 200 | $-1.489\pm0.065$† | $< 10^{-10}$ | $< 10^{-10}$ |
| | | RBM | 200 | $-1.530\pm0.105$† | $< 10^{-10}$ | $< 10^{-10}$ |
| | | B1 | 200 | $-1.335\pm0.111$† | $< 10^{-10}$ | $< 10^{-10}$ |
| | Price Luxury$\rightarrow$Budget | Proposed | 100 | $-0.088\pm0.090$* | $1.5\times10^{-3}$ | 0.039 |
| | | RBM | 100 | $+0.042\pm0.024^{ns}$ | 0.195 | 0.747 |
| | | B1 | 100 | $-0.024\pm0.012^{ns}$ | 0.511 | 0.853 |
| | Brand MAC$\rightarrow$L'Oréal | Proposed | 100 | $+0.017\pm0.036^{ns}$ | 0.545 | 0.382 |
| | | RBM | 100 | $-0.034\pm0.033^{ns}$ | 0.205 | 0.346 |
| | | B1 | 100 | $+0.020\pm0.072^{ns}$ | 0.564 | 0.778 |

## 7.2 Coupling-aware combined clamp sharpens distributional match

We further test whether jointly clamping a coupled pair (Rating $= 1$ *and* Topic(SERVICE)$= 1$) brings the generated distribution closer to the model's naturally-conditioned low-rating generations than clamping Rating alone. We report Kolmogorov–Smirnov $p$ and Wasserstein distance for two reference conditionings: all reviews with Rating$= 1$, and the conditional subset Rating$= 1\cap$Service; in both cases the reference distribution is the model's naturally-conditioned generations on those samples, not the review texts themselves. Full KS/Wasserstein statistics for all three models are given in Table 9 (App. H).

Under the all-R$=1$ reference the combined clamp reduces Wasserstein by 11% (Smartphone, $0.235 \rightarrow 0.210$) and 23% (Beauty, $0.147 \rightarrow 0.113$); the effect is Beauty-dominant. The sharpening is specific to this model-

---

[1]CTRL and PPLM are omitted here: their form-side outputs lie outside the valid review distribution (Table 1) so pre/post-clamp $\Delta$s would measure OOD shifts, and they implement attribute control via marginal channels rather than the joint-distribution exploitation the structure-level tests require.

consistent reference: against the natural review *texts* the same change yields no improvement on Beauty (Table 6, §7.4). Figure 2 visualizes the mechanism: the natural low-rating mass concentrates near $-1$, the rating-only clamp reproduces this mode while retaining residual positive mass, and the combined clamp moves the residual toward the natural Rating= 1∩Service profile. The RBM reproduces the sharpening more strongly ($-29\%/-62\%$: $0.319 \rightarrow 0.227$, $0.110 \rightarrow 0.042$) and on Beauty is the only model whose combined-clamp match is KS-rejected ($p=0.286$): higher-order propagation of a coupled clamp does not require depth. B1 also improves from rating-only to combined, but matches the reference far more loosely than either energy-based model in every cell, with no KS non-rejection anywhere: re-conditioning shifts the mean, while clamping matches the shape.

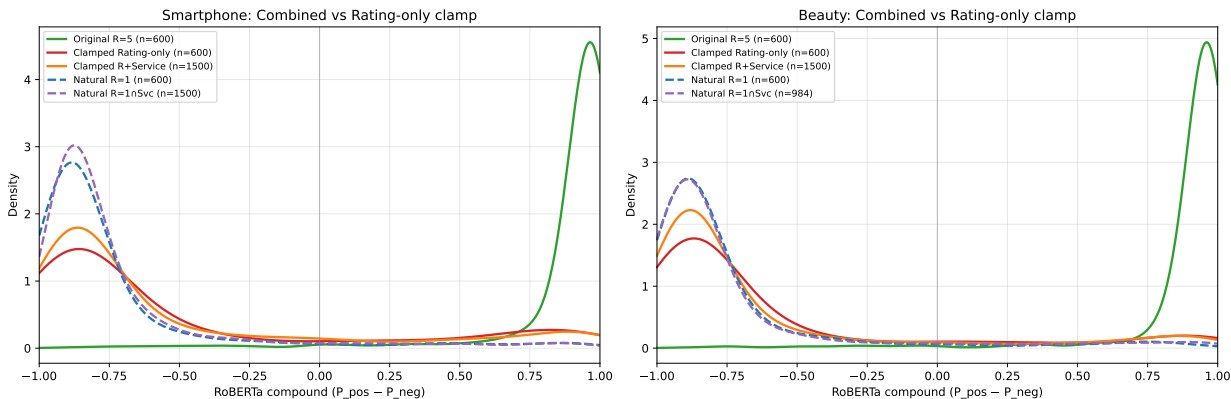

Figure 2: Combined clamp KDE (RoBERTa). The single-attribute Rating clamp (red) approximates the natural Rating= 1 distribution (blue, dashed) but retains residual positive mass; the coupling-aware combined clamp (orange) tightens the match toward the natural Rating= 1∩Service mode (purple, dashed).

### 7.3 Content-axis clamp via brand–topic specialty

Utilizing Beauty's domain-specific Brand–Topic specialty couplings (Sally Hansen–Nails, Bath & Body Works–Fragrance), we perform content-axis clamps. For each pair, we clamp the brand and target topic visible units and generate $n=500$ samples per seed under the 3-seed evaluation, then measure: keyword frequency (average per-generation count of a hand-curated topic lexicon under case-insensitive whole-word match[2]), SBERT similarity to natural baseline (mean-best-match cosine), and sentiment (to verify content/sentiment independence).

Keyword frequency rises sharply in both pairs (Table 5)—roughly 3× over baseline for Sally Hansen → Nails and 21× for Bath & Body Works → Fragrance—and the DBM-mediated shift is 1.9–3.4× B1's under the same clamp; the SBERT pull toward the target Brand∩Topic mode is likewise 1.6–3.0× B1's. The RBM lands between the two, a consistent DBM > RBM > B1 staircase in content strength. Orthogonality is where depth becomes load-bearing: the RBM's content clamp leaks significantly into sentiment on Sally Hansen where the DBM remains null—a weaker form of the entanglement failure that §7.4 documents for activation steering. The sentiment-axis control test is otherwise largely respected: the DBM is null on Sally Hansen × Nails, and the small positive shift on Bath & Body Works × Fragrance appears equally in B1, marking it as a domain artifact (fragrance reviews are intrinsically positive-leaning) rather than a handle-specific leak. The two-axis decomposition thus holds in the strong sense for Sally Hansen and in a qualified sense for Bath & Body Works, where the residual is an order of magnitude smaller than the Rating effect ($|\Delta| \geq 0.9$). App. N provides representative verbatim samples for each axis.

---

[2]Nails: {nail, polish, color, manicure, pedicure, lacquer, enamel}. Fragrance: {fragrance, scent, smell, perfume, aroma, cologne}.

Table 5: Brand–Topic specialty clamp (Beauty), 3-seed mean ± std. kw orig→int: topic-keyword frequency of the original vs. clamped generations; kw nat.: keyword frequency of the model's naturally-conditioned generations on the natural Brand∩Topic samples ($n$=51/70); SBERT$_{o \to b}$/SBERT$_{i \to b}$: mean-best-match cosine of original/clamped generations to those same generations (model-consistent reference; higher = closer to the target mode). †: $p < 10^{-3}$; ∗: $p < 0.05$; n.s.: $p \geq 0.1$ (item-level paired-$t$ on ΔRoBERTa). RBM rows omit the reference-based columns, which require the model's own naturally-conditioned generations.

| Pair | ΔRoBERTa | kw orig→int | kw nat. | SBERT$_{o \to b}$ | SBERT$_{i \to b}$ |
|---|---|---|---|---|---|
| Sally Hansen × Nails | | | | | |
| Proposed (DBM + adapter) | −0.015±0.012 | 0.29 → 0.86 | 4.08 | 0.398±0.024 | 0.458±0.017 |
| RBM ablation | −0.048±0.022 † | 0.24 → 0.61 | — | — | — |
| B1: Direct MLP (no DBM) | −0.002±0.021 | 0.26 → 0.44 | 4.05 | 0.412±0.010 | 0.431±0.011 |
| B&B Works × Fragrance | | | | | |
| Proposed (DBM + adapter) | +0.046±0.012 * | 0.03 → 0.63 | 2.93 | 0.430±0.012 | 0.506±0.004 |
| RBM ablation | +0.020±0.046 | 0.04 → 0.48 | — | — | — |
| B1: Direct MLP (no DBM) | +0.048±0.026 * | 0.03 → 0.35 | 2.37 | 0.448±0.008 | 0.496±0.012 |

## 7.4 Comparison with activation steering

We finally implement CAA on the same frozen Qwen2.5-0.5B-Instruct generator: a difference-of-means steering vector (natural Rating=1 vs. Rating=5 reviews) is added to one decoder block during decoding, with (layer, $\alpha$) selected *in CAA's favor* on held-out dev prompts by maximizing the sentiment shift under a fluency guard; the combined condition adds the sum of the rating and Topic(SERVICE) vectors, CAA's native vector-arithmetic composition (full setup in App. G). Only the generation seed varies across the 3-seed evaluation, as for the other non-trainable baselines.

**Sentiment axis: steering is a competent handle, but not distributionally natural.** On Rating 5→1 CAA produces ΔRoBERTa=−1.011/−1.290 (Smartphone/Beauty) against the proposed model's −1.388/−1.489 and B1's −1.279/−1.335 (Table 4). A single correlational direction therefore suffices to move the *mean* of the sentiment distribution by an amount comparable to clamping, as the steering literature predicts. The model-consistent reference of Table 9 is undefined for CAA: unsteered, the system has no conditioning channel and its generations on natural low-rating prompts remain strongly positive, leaving no naturally-conditioned generation to compare against. We therefore score every system against the same model-independent reference: the sentiment distribution of the natural review *texts* (Rating=1 for rating-only; Rating=1∩Service for combined). Table 6 shows that the proposed model attains the best Wasserstein in three of four cells and is the only system with any KS non-rejections (its own Smartphone combined cell is rejected at $p$=0.004); CAA is rejected in every cell ($p$≈0), including Beauty rating-only where its Wasserstein is nominally smallest—the first moment matches, the shape does not. Diagnostically, the combined condition *degrades* CAA on Smartphone ($W$: 0.347 → 0.448): summed steering vectors interfere, while coupling-aware clamping improves the match in both domains.

Table 6: Distribution match against the natural review-text reference (RoBERTa), 3-seed mean ± std. Wasserstein distance (smaller is better) and KS $p$ (higher = harder to distinguish from the natural texts; 0.000 = rejected).

| | | Proposed | | B1 | | CAA | |
|---|---|---|---|---|---|---|---|
| Domain | Condition | $W$ | KS $p$ | $W$ | KS $p$ | $W$ | KS $p$ |
| Smartphone | rating-only | **0.216 ± 0.063** | 0.098 ± 0.122 | 0.317 ± 0.075 | 0.002 | 0.347 ± 0.017 | 0.000 |
| Smartphone | combined | **0.193 ± 0.046** | 0.004 ± 0.006 | 0.283 ± 0.059 | 0.000 | 0.448 ± 0.003 | 0.000 |
| Beauty | rating-only | 0.124 ± 0.063 | **0.374 ± 0.325** | 0.285 ± 0.097 | 0.013 | **0.101 ± 0.013** | 0.000 |
| Beauty | combined | **0.138 ± 0.078** | 0.096 ± 0.069 | 0.248 ± 0.039 | 0.000 | 0.247 ± 0.008 | 0.000 |

**Content axis: steering moves keywords, not controlled content.** We construct brand and topic steering vectors for the two Brand–Topic specialty pairs and add their sum, re-tuned in CAA's favor on a keyword-shift objective (App. G). Against the model-independent reference (Table 7), steering moves target keywords far more aggressively than clamping ($0.29 \to 2.12$, $0.09 \to 4.31$, the latter overshooting the natural rate of 1.30 by $3.3\times$), but the shift is not controlled content. Two-axis independence fails: the Sally Hansen content edit drags sentiment by $\Delta_{\mathrm{RoBERTa}} = -0.605$ (proposed: $-0.015$, n.s.), and Bath & Body Works leaks $+0.127$ ($2.8\times$ the proposed model's residual). And on Sally Hansen the steered generations move *away* from the natural target mode (SBERT $0.384 \to 0.278$) where clamping moves toward it ($+0.060$). The mechanism is the contrast sets' incidental valence skews ($-0.02/-0.03$ Nails/Sally Hansen; $+0.13/+0.06$ Fragrance/Bath & Body Works), whose signs match the induced sentiment shifts: a difference-of-means direction carries every co-varying attribute, and the coefficient required for a content-sized effect amplifies the leak. The DBM clamp does not inherit this entanglement, because mean-field inference propagates the clamp through a joint distribution in which Rating and Topic are distinct variables.

Table 7: Content-axis clamp (Brand×Topic clamp vs. brand+topic steering vector) against the model-independent reference: the natural Brand∩Topic review texts ($n$=51/70; keyword rates 1.94/1.30). 3-seed mean $\pm$ std. $\Delta$RoBERTa: sentiment side-effect (should be $\approx 0$ for a controlled content edit); SBERT$_{o \to n}$/SBERT$_{i \to n}$: mean-best-match cosine of original/clamped generations to the natural texts. For Proposed/B1 the deltas are nearly identical to the model-consistent ones of Table 5.

| Pair | Model | $\Delta$RoBERTa | kw orig→int | SBERT$_{o \to n}$ | SBERT$_{i \to n}$ |
|---|---|---|---|---|---|
| Sally Hansen × Nails | Proposed | $-0.015 \pm 0.012^{ns}$ | $0.29 \to 0.86$ | $0.373 \pm 0.005$ | $\mathbf{0.433 \pm 0.005}$ |
| | B1 | $-0.002 \pm 0.021^{ns}$ | $0.26 \to 0.44$ | $0.370 \pm 0.002$ | $0.394 \pm 0.006$ |
| | CAA | $-0.605 \pm 0.041\,\dagger$ | $0.29 \to 2.12$ | $0.384 \pm 0.001$ | $0.278 \pm 0.003$ |
| B&B Works × Fragrance | Proposed | $+0.046 \pm 0.012\,*$ | $0.03 \to 0.63$ | $0.412 \pm 0.004$ | $0.484 \pm 0.010$ |
| | B1 | $+0.048 \pm 0.026\,*$ | $0.03 \to 0.35$ | $0.408 \pm 0.003$ | $0.462 \pm 0.009$ |
| | CAA | $+0.127 \pm 0.031\,\dagger$ | $0.09 \to 4.31$ | $0.438 \pm 0.002$ | $\mathbf{0.644 \pm 0.001}$ |

$\dagger$: item-level paired-$t$ $p < 10^{-3}$; *: $p < 0.05$; ns: $p \geq 0.1$.

From the results so far, CAA is the strongest competing handle we evaluate, and it differs from the proposed architecture in kind, not degree: it shifts sentiment means but is everywhere rejected on distributional match to natural reviews under RoBERTa, composes destructively where coupling-aware clamps sharpen, entangles sentiment on the content axis, and requires a hand-built contrast set plus a per-task (layer, $\alpha$) sweep—whereas both axes of §7.1–§7.3 are clamps on a single learnt belief space that also exposes a coherence score (§6) and discovered coupling structure to exploit. Steering edits activations inside the generator; clamping consults an explicit joint model outside it.

## 8 Discussion and Conclusion

The architectural split introduced in §1 generated a clear two-sided prediction, and both sides are fully confirmed by the results. On the form side, Direct-MLP (B1), in which the MLP receives the LLM's CE gradient all the way back to the inputs while the DBM's training signal is blocked at the adapter boundary, matches or marginally outperforms the proposed model (Table 1, App. C). Both approaches substantially outperform LoRA, full fine-tuning, CTRL, PPLM, and one-shot ICL. We attribute this shared advantage to the soft-prompt channel itself (Lester et al., 2021).

On the structure side, the proposed model outperforms B1 across every comparison: its energy responses faithfully track each domain's empirical coupling structure, its clamps shift sentiment and content selectively while composing cleanly along coupled attributes, and the two control handles remain nearly orthogonal. By contrast, a generously tuned CAA baseline matches raw sentiment shifts but fails to preserve distributional consistency, exhibits destructive composition, and entangles the control axes (§7.4). B1 can be re-conditioned but cannot be clamped, as it lacks an explicit belief space over which to factor. Consequently, our central claim is established not by denying B1 its form-level advantage, but by demonstrating that high form quality and effective structure-level control constitute distinct axes of system design.

The ablations locate where each architectural ingredient is load-bearing. An explicit energy model is indispensable for coherence scoring and clamp-and-re-equilibrate generation: the correlation baseline recovers the coupling ranking but provides neither an energy function nor a generative path. A latent joint-density model of any depth then suffices for high form quality, accurate coupling recovery, effective sentiment-axis clamping, and distributional sharpening under combined clamps—on this last metric the single-layer RBM even matches or exceeds the DBM. Hierarchical latent structure, however, is essential for the content axis: the DBM produces consistently stronger content shifts and, crucially, prevents content clamps from leaking into sentiment. In a single-layer hidden space, brand, topic, and rating collapse into a shared representation, such that editing content drags sentiment along with it. In short, depth enables independent multi-attribute control: the core capability this framework is designed to provide. App. J conceptually extends this stratification to other explicit joint families (Ising/log-linear models, Bayesian networks, VAEs, and masked-feature predictors).

We draw two conclusions for practice. First, when methods such as LoRA already approach the ceiling on cross-entropy and embedding similarity, further marginal gains on those metrics no longer discriminate among architectures; the discriminative axis is whether a system supports coherence scoring and predictively valid attribute clamps. Second, interpretability and controllability should be first-class architectural requirements rather than post-hoc probes on monolithic models. The present architecture is one concrete realization, not a unique one. The mouth speaks; the brain, separately and inspectably, decides what to say.

**Scope and limitations. Causal interpretation.** "Clamping" here denotes fixing visible units in the learnt DBM and re-running mean-field inference before decoding; the selectivity reflects *model-internal distributional consistency*, not identification of real-world causal effects. We do not specify a structural causal model nor address latent confounding. **Form/meaning framing.** We adopt Bender & Koller (2020)'s form/meaning distinction primarily as an *organizing framework*. This does not entail endorsement of their stronger claim that LLMs trained on linguistic form alone cannot acquire meaning. Rather, our central claim is that form-level metrics alone are insufficient to evaluate structure-level capabilities, which require structural clamping. **Sample sizes.** Exp I uses $n=1024$/seed; Exp III uses $n=200$/seed for Rating and $n=100$/seed for Price/Brand. Seed-to-seed std is $\leq 0.009$ on CE/CosSim and $\leq 0.09$ on clamp $\Delta$. **Cross-model and cross-domain scope.** While transfer to Llama-3.2-1B-Instruct and OLMo-2-1B-Instruct is validated under the full 3-seed protocol (App. K), broader generalization (e.g., other categories, languages, non-review text, larger generators) remains to be verified.

**Reproducibility.** Code and pre-trained model checkpoints will be made publicly available upon acceptance.

## Broader Impact Statement

This work concerns controllable generation of synthetic consumer reviews and therefore carries a direct dual-use risk. The legitimate applications we target are market simulation, data augmentation for structured review domains, and preference modeling over explicitly represented attributes. The same controllability, however, could support deceptive product promotion, fabricated negative reviews against competitors, reputation manipulation, or large-scale opinion shaping. Two considerations bound the marginal risk. First, fluent fake reviews are already cheap: any instruction-tuned LLM produces them from a short prompt, and our 0.5B generator adds no fluency beyond that baseline. What this architecture adds is distribution-level control over attribute configurations — a meaningful increment mainly for systematic, large-scale manipulation rather than for individual fake reviews. Second, the explicitness of the attribute channel cuts both ways: because every generation is conditioned on an inspectable attribute configuration, synthetic corpora produced by this pipeline are auditable in a way that prompt-level manipulation is not; the conditioning can be logged, disclosed, and inspected. Appropriate safeguards for deployment include disclosure of synthetic origin, machine-generated-text detection and watermarking, and platform-level provenance requirements for review content. Our experiments use the public Amazon Reviews dataset (McAuley et al., 2015; He & McAuley, 2016); visible features are derived from review text and metadata already released in that dataset and contain no additional personally identifying information. Released code and checkpoints are intended for research on controllable generation and simulation, not for producing deployable review content.

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

## A  Mean-Field Energy (Coherence Score)

The coherence score reported throughout the paper as "$\tilde{F}(v)$" is the expected energy under the converged Bernoulli mean-field distribution $q(h) = \prod_{l,i} \text{Bern}(\mu_i^{(l)})$:

$$\tilde{F}(v) \;=\; \mathbb{E}_q[E(v,h)] \;=\; -\sum_{l=1}^{L} \mu^{(l-1)\top} W^{(l)} \mu^{(l)} - b^\top v - \sum_{l=1}^{L} c^{(l)\top} \mu^{(l)}, \tag{4}$$

where $\mu^{(0)} \equiv v$. This is the same quantity used in the positive/negative-phase terms of PCD training (`free_energy()` in the implementation), kept consistent between training and evaluation. It is an upper bound on the true variational free energy by exactly $H(q)$, the entropy of the mean-field distribution. The entropy term is omitted because (i) this is the canonical PCD quantity, kept consistent between training and evaluation, and (ii) empirically, including it does not change the conclusions: recomputing the interaction-contrast analysis of §6 with the full variational free energy $\tilde{F}(\mathbf{v}) - H(q)$ yields Pearson $+0.928 \pm 0.009$ (Smartphone) and $+0.908 \pm 0.010$ (Beauty) against the empirical coupling, versus $+0.941/+0.802$ for the expected energy (permutation $p=2 \times 10^{-4}$ in all cases), with per-pair responses correlated $+0.95/+0.93$ between the two scorings.

## B  Text Prompt

The same template is used in both domains, with `{domain}` substituted for the noun of the product category (e.g., `smartphone`, `beauty product`) shown in Fig. 3.

## C  Direct-MLP Baseline: Structural Argument for the CE Tie

In Table 1, under 3-seed evaluation (item-level inference, §5) Direct-MLP (B1) ties the proposed model on Smartphone CE ($p=0.383$) and Beauty CosSim ($p=0.354$), and resolves the other two cells in its own favor: Smartphone CosSim ($p=1.3\times10^{-3}$) and Beauty ($p=2.9\times10^{-3}$), both within the seed-to-seed std. This appendix explains why CE parity (and the small tilt in B1's favor) is structurally expected and what it does (and does not) imply about architectural separation.

**Why the tie is structurally inevitable, not a counter-argument.**  B1 has a built-in advantage on the cross-entropy metric. Its MLP is trained end-to-end on the LLM's CE loss with full access to the visible features $v$: the gradient flows from the LLM token loss through the soft prompt all the way back to the

```
### Instruction:
You are a helpful assistant to generate online review about {domain} in Amazon.
Consider product name, price in dollars, and average rating.

### Example:
Product name: {example_name}
Price: ${example_price}
Average rating: {example_rating}
Review: {example_review}

### Task:
Product name: {product_name}
Price: ${product_price}
Average rating: {average_rating}
Review:
```

Figure 3: Prompt template

visible-input MLP. The proposed model's DBM, by contrast, is optimized for the joint visible likelihood $P(v)$ via persistent contrastive divergence; the LLM's CE gradient is blocked at the adapter boundary, so the DBM never sees CE at all. Asymptotically, given enough adapter capacity and enough data, the end-to-end-CE-optimized B1 is therefore expected to converge to a CE optimum the proposed model cannot exceed by definition. CE parity at scale is the predicted regime, not a refutation.

As shown in the main text, the architectural separation principle is established not by surface metrics but by the capabilities the DBM exposes more directly than an MLP-based projection: energy-based coherence scoring over arbitrary configurations (§6), clamp-based control with predictively valid couplings, and two-axis independent clamp-based control (§7). B1 can be re-conditioned by hand-edited inputs, but it has no scalar coherence score over arbitrary configurations, no closed-form mean-field semantics for clamping a subset of the visible units, and no joint-likelihood-trained latent space whose marginals can be clamped upon.

## D  Preprocessing Details

Raw data come from Amazon Reviews 2023 (McAuley et al., 2015) (review and product-metadata files for *Cell Phones and Accessories* and *All Beauty*). Products are filtered by main category ("Phones"/"Beauty") and a minimum price ($100 for Smartphone, $5 for Beauty); for Smartphone, accessories are excluded via category sets and title keywords (e.g., cases, cables, chargers). Reviews are restricted to verified purchases, merged with product metadata by parent ASIN, filtered by character length, and restricted to English via FastText (Bojanowski et al., 2017; Joulin et al., 2017)) language identification (`lid.176`; §4). One review per user is retained (the highest-priced purchase). Binary features for the input are then constructed as one-hot tags as follows:

**Brand:** a curated per-domain brand vocabulary matched case-insensitively against the product title, then the store name, with aliases mapped to canonical brands (e.g., iPhone→Apple, Galaxy→Samsung) and unmatched products assigned to *Others*.

**Price:** the product price binned into four tiers (Smartphone: Entry/Mid/High/Premium with edges $150/$400/$800; Beauty: Budget/Mid/Premium/Luxury with edges $10/$25/$50).

**Rating:** one-hot integer star rating $\in \{1, 2, 3, 4, 5\}$.

**Topic:** hand-curated keyword lexicons (22 topics for Smartphone, 17 for Beauty) matched case-insensitively against the review text.

**Purchase:** statistics of the user's full within-category review history: purchase-frequency bins (1, 2–4, $\geq 5$, $\geq 10$), maximum-price thresholds ($\geq \$500$, $\geq \$1000$), a minimum-price threshold ($\geq \$100$), and an Amazon-Renewed store flag.

**PriceRange:** bins of the user's max/min purchase-price ratio ($<10$, 10–50, 50–100, $\geq 100$).

**Auto:** the top-100 terms using term frequency inverse document frequency (TF-IDF; Sparck Jones, 1972), binarized as presence indicators (alphabetic tokens of $\geq 3$ characters, document frequency in $[0.005, 0.7]$, English plus domain-specific stop-word lists).

The resulting tagged table yields the $J_v$=160/170 visible vectors and is split into train/val/test = 52,952/1,024/1,024 per domain with a fixed split seed (§4).

## E   Training Hyperparameters

Table 8 lists the full hyperparameter settings used in the three training phases: layer-wise pretraining (LPT), joint finetuning (JFT), and adapter training. Values are held identical across both domains to keep the cross-domain comparison consistent.

Table 8: Training hyperparameters for the three-phase procedure. The adapter uses gradient accumulation over 4 steps (effective batch 64).

| Parameter | LPT | JFT | Adapter |
|---|---|---|---|
| Epochs | 500 | 300 | 10 |
| Batch size | 512 | 512 | 16×4 |
| Learning rate | 0.01 | 0.001 | 0.01 |
| Weight decay | - | $10^{-3}$ | $10^{-4}$ |
| CD/PCD steps | 5 | 5 | - |
| Soft tokens size $K$ | - | - | 16 |
| Mean-field iterations | - | 10 | - |

The DBM uses hidden layers $[512, 1024, 2048]$ (2.71M energy-model parameters for $J_v$=160/170); the RBM ablation uses a single hidden layer of 3584 units ($\approx$0.58M). The adapter is identical in both cases.

## F   Depth Ablation with RBM: Setup and Energy Scoring

**Setup.** The RBM ablation replaces the $[J_v, 512, 1024, 2048]$ DBM with a single-hidden-layer model $[J_v, 3584]$: the total hidden dimensionality (512+1024+2048=3584) is preserved, so the concatenated belief vector $\mathbf{H}$ has identical dimension and the adapter is architecturally identical (i.e., only latent depth is ablated). Attribute model parameters are $\approx$0.58M (RBM) and 2.71M (DBM). A single-hidden-layer model is mathematically an $L$=1 DBM, so training uses the identical pipeline and seeds $\{0, 1, 2\}$ (for $L$=1, joint fine-tuning coincides with standard PCD training of an RBM); the adapter is trained with the identical protocol of App. E.

**Scoring must match the inference scheme.** For coherence scoring, the single-layer model admits exact marginalization over its hidden units,

$$F(\mathbf{v}) \;=\; -\mathbf{b}^\top \mathbf{v} \;-\; \sum_i \mathrm{softplus}\big(W_{:,i}^\top \mathbf{v} + c_i\big), \tag{5}$$

its exact free energy. Scoring the RBM instead with the deep model's mean-field variational energy (App. A) produces a false negative on the coupling analysis of §6: the $r$-association collapses to $-0.192 \pm 0.051$ (Smartphone; permutation $p$=0.38) and $-0.088 \pm 0.072$ (Beauty; $p$=0.73), versus $+0.912 \pm 0.002$ and $+0.724 \pm 0.079$ under exact scoring. Energy-based coherence scores are therefore comparable across models only when each model is scored under the energy consistent with its own inference scheme: the mean-field expected energy for the DBM (App. A) and the exact free energy for the RBM.

## G   CAA Steering Baseline: Setup

The rating steering vector is the unit-normalized difference of mean residual-stream activations between natural Rating=1 and Rating=5 reviews (256 per side, disjoint from the generation samples), added to one decoder block during decoding with coefficient $\alpha$; original and steered generations share an identical text prompt, so the activation edit is the only modification channel. The sentiment-axis hyperparameters (block 18 of 24, $\alpha$=16, per domain) are selected in CAA's favor on held-out dev prompts by maximizing the sentiment shift over a $4 \times 3$ (layer, $\alpha$) grid, subject to a fluency guard (median steered perplexity $\leq 3\times$ unsteered; the selected configurations show no degradation). For the content axis (§7.4), brand and topic vectors are constructed analogously and (layer, $\alpha$) is re-tuned in CAA's favor on a keyword-shift objective under the same fluency guard, selecting block 14, $\alpha$=16. Extraction is deterministic, so only the generation seed varies across the 3-seed evaluation.

## H   Combined Clamp: Full Distribution Statistics

Table 9 reports the full Kolmogorov–Smirnov (KS) and Wasserstein statistics for the combined-clamp analysis of §7.2, for all three clampable models under both model-consistent reference conditionings. Each block contrasts the rating-only clamp against the combined (Rating $\cap$ Service) clamp; higher KS $p$ means the clamped distribution is harder to distinguish from the reference, and lower $W$ means a closer match. Note that the reference is *model-specific* (each model's own naturally-conditioned generations), so rows compare how faithfully each model reproduces its own natural conditioning, not distance to a shared external target; the shared-target comparison against natural review texts is Table 6.

Three patterns summarize the table. (i) For the proposed model, the combined clamp improves the match over rating-only under the all-R=1 reference ($0.235 \rightarrow 0.210$ Smartphone, $0.147 \rightarrow 0.113$ Beauty). (ii) The RBM reproduces this sharpening more strongly ($0.319 \rightarrow 0.227$, $0.110 \rightarrow 0.042$) and is the only model whose Beauty combined cells are not KS-rejected ($p$=0.286/0.311): higher-order propagation of a coupled clamp does not require depth. (iii) B1 also improves from rating-only to combined—it receives both edited attributes as inputs—but its match is the loosest of the three models in every Beauty cell and it reaches no KS non-rejection anywhere, consistent with the reading that re-conditioning shifts the mean of the distribution while clamping matches its shape.

## I   Multi-Classifier Validation of Clamp Effects

To verify that the clamp effects of §7 are not artifacts of the RoBERTa scorer, we re-score the generated texts with two independent classifiers: a review-domain star-rating classifier (`nlptown/bert-base-multilingual-uncased-sentiment`; we report the expected star rating $\sum_k k\,p_k \in [1,5]$) and a zero-shot topic classifier (`facebook/bart-large-mnli`; we report the entailment score of the clamp-target topic label). Table 10 reports 3-seed means.

Both validations reproduce the findings of §7. On the Rating $5 \rightarrow 1$ clamp, every system moves the expected star rating from $\sim$4.4–4.7 to $\sim$1.9–2.2, with the same ordering as the RoBERTa analysis; CAA's unsteered generations start lower ($\sim$3.8) because it has no conditioning channel. On the content axis, the DBM > RBM > B1 staircase of Table 5 replicates under the independent topic classifier (target-topic gains $+0.13 > +0.07 > +0.03$ on Sally Hansen–Nails; $+0.26 > +0.21 > +0.16$ on Bath & Body Works–Fragrance), and CAA's behavior again splits by pair exactly as in §7.4: a small target-topic gain on Sally Hansen (where its edit leaks into sentiment instead) and a runaway shift to 0.93 on Fragrance—roughly twice the DBM's clamped endpoint—mirroring the keyword overshoot of Table 7.

## J   Other Explicit Joint Attribute Models

The capability stratification of §8 extends conceptually to the other explicit joint-model families along three axes: (i) does the family expose a scalar coherence score for arbitrary full attribute configurations, (ii) does

Table 9: Coupling-aware combined clamp vs. single-attribute Rating clamp under 3-seed evaluation (mean ± std) using RoBERTa sentiment. Distance to the model's naturally-conditioned low-rating generations (the *model-consistent* reference: generation conditioned on natural Rating=1 / Rating=1∩Service samples without clamp); smaller is better.

| Domain | Comparison | | | Model | KS $p$ | Wasserstein |
|---|---|---|---|---|---|---|
| | rating-only | ↔ | all R=1 | Proposed | $0.006 \pm 0.008$ | $0.235 \pm 0.054$ |
| | | ↔ | | RBM | $0.004 \pm 0.006$ | $0.319 \pm 0.106$ |
| | | ↔ | | B1 | $0.015 \pm 0.017$ | $0.269 \pm 0.077$ |
| Smartphone | combined | ↔ | all R=1 | Proposed | $0.000 \pm 0.000$ | $0.210 \pm 0.066$ |
| | | ↔ | | RBM | $0.021 \pm 0.027$ | $0.227 \pm 0.075$ |
| | | ↔ | | B1 | $0.013 \pm 0.018$ | $0.232 \pm 0.053$ |
| | combined | ↔ | R=1∩Service | Proposed | $0.017 \pm 0.023$ | $0.192 \pm 0.064$ |
| | | ↔ | | RBM | $0.001 \pm 0.001$ | $0.247 \pm 0.073$ |
| | | ↔ | | B1 | $0.000 \pm 0.001$ | $0.250 \pm 0.058$ |
| | rating-only | ↔ | all R=1 | Proposed | $0.013 \pm 0.012$ | $0.147 \pm 0.030$ |
| | | ↔ | | RBM | $0.082 \pm 0.069$ | $0.110 \pm 0.060$ |
| | | ↔ | | B1 | $0.003 \pm 0.004$ | $0.249 \pm 0.058$ |
| Beauty | combined | ↔ | all R=1 | Proposed | $0.078 \pm 0.100$ | $0.113 \pm 0.016$ |
| | | ↔ | | RBM | $0.286 \pm 0.211$ | $0.042 \pm 0.025$ |
| | | ↔ | | B1 | $0.030 \pm 0.022$ | $0.163 \pm 0.007$ |
| | combined | ↔ | R=1∩Service | Proposed | $0.028 \pm 0.036$ | $0.091 \pm 0.020$ |
| | | ↔ | | RBM | $0.311 \pm 0.389$ | $0.062 \pm 0.050$ |
| | | ↔ | | B1 | $0.016 \pm 0.020$ | $0.181 \pm 0.051$ |

Table 10: Multi-classifier validation, 3-seed mean of per-seed means (original → clamped). Left: expected star rating under the review-domain classifier (Rating $5 \to 1$ clamp, $n$=200/seed). Right: zero-shot target-topic score (content clamp, $n$=500/seed). Seed-to-seed std $\leq 0.19$ stars and $\leq 0.05$ topic score in every cell; full statistics in the repository (`classifier_rescore_summary.csv`).

| Model | Expected stars (Rating $5 \to 1$) | | Target-topic score (content) | |
|---|---|---|---|---|
| | Smartphone | Beauty | SH × Nails | B&BW × Fragrance |
| Proposed | $4.48 \to 1.93$ | $4.65 \to 1.96$ | $0.07 \to 0.20$ | $0.19 \to 0.45$ |
| RBM | $4.43 \to 2.21$ | $4.67 \to 1.93$ | $0.07 \to 0.14$ | $0.18 \to 0.40$ |
| B1 | $4.42 \to 2.10$ | $4.63 \to 2.25$ | $0.06 \to 0.09$ | $0.17 \to 0.33$ |
| CAA | $3.76 \to 2.13$ | $3.84 \to 1.97$ | $0.06 \to 0.10$ | $0.18 \to 0.93$ |

it support clamp-and-re-equilibrate semantics over the remaining attributes, and (iii) does it expose a latent state that can serve as the adapter's conditioning interface?

**Ising/log-linear models** (fully visible pairwise MRFs) define an explicit energy, so they support (i) and, via Gibbs updates on the visibles, (ii); trained by pseudo-likelihood they are the cheapest member of the energy family. What they lack is (iii): with no hidden units there is no belief state to project into soft prompts, so they cannot condition the generator in this architecture, and pairwise sufficient statistics bound the interaction order they can represent—the higher-order brand–topic–rating structure that hierarchical latents separate (§7.3) is outside the family. **Bayesian networks** support principled conditioning (ii) and exact or approximate likelihoods (i), but require committing to a directed structure over attributes that have no natural ordering in co-occurrence data; with latent nodes they reacquire the inference machinery of undirected models while keeping the structure-selection problem. **VAEs** provide a continuous latent state (iii) that could feed an adapter directly, but no energy function: coherence scoring must proceed through an ELBO estimate (i, weakly), and clamping a subset of observed attributes has no native re-equilibration semantics—posterior inference over the unclamped attributes requires auxiliary machinery (ii,

weakly). **Masked-feature predictors** answer conditional queries $P(\text{masked} \mid \text{rest})$ directly, which suffices for coupling discovery (cf. the correlation baseline of §6), but define no joint energy and no equilibrium state; composing several simultaneous clamps is heuristic rather than inferential.

The DBM occupies the corner of this space where all three properties coincide: an explicit energy over arbitrary configurations, mean-field clamp-and-re-equilibrate inference, and a latent belief state that doubles as the conditioning interface.

## K   Proposed Model under Alternative LLM Backbones

To verify that the proposed pipeline (DBM → adapter → frozen generator) is generator-agnostic, we re-evaluate the proposed model under the full 3-seed protocol, swapping only the frozen language model and retraining only the adapter (the soft-prompt embedding dimension changes with the generator's hidden size). Unlike the main experiments, the DBM checkpoint is held fixed across seeds and backbones (one per domain), so the seeds vary only adapter initialization, data shuffling, and decoding: the protocol directly measures the robustness of adapter-only transfer, which is the claim under test. We compare two 1B-scale families spanning independent organizations and tokenizers against the main-text reference: Qwen2.5-0.5B-Instruct (Alibaba), Llama-3.2-1B-Instruct (Meta), and OLMo-2-0425-1B-Instruct (AI2).

Table 11: Cross-LLM validation of the proposed model, 3-seed mean $\pm$ std. Same DBM checkpoint; only the adapter is retrained for each backbone. CE on $n$=1024, CosSim on $n$=500 per seed; the Qwen row restates the main-text 3-seed values (Table 1; CosSim there on $n$=1024).

| Frozen generator | Params | Smartphone | | Beauty | |
|---|---|---|---|---|---|
| | | CE | CosSim | CE | CosSim |
| Qwen/Qwen2.5-0.5B-Instruct (main) † | 0.5B | 3.135±0.005 | 0.534±0.007 | 3.181±0.006 | 0.455±0.005 |
| meta-llama/Llama-3.2-1B-Instruct † | 1B | 2.954±0.004 | 0.496±0.003 | 2.946±0.001 | 0.446±0.009 |
| allenai/OLMo-2-0425-1B-Instruct † | 1B | **2.854±0.001** | **0.543±0.004** | **2.878±0.003** | **0.475±0.004** |

The pipeline transfers without modification across all three families, and the transfer is robust to seed: seed-to-seed std is $\leq$0.004 on CE and $\leq$0.009 on CosSim in every cell. Cross-entropy improves with generator scale: Qwen-0.5B's CE near 3.1 drops to roughly 2.9 on both 1B-class generators, with OLMo-2-1B-Instruct lowest in both domains (2.854 Smartphone, 2.878 Beauty); both CE improvements over the Qwen reference are significant (item-level paired $t$ on the shared test items, $p < 10^{-10}$ in all four cells). SBERT cosine similarity behaves less monotonically and is sensitive to tokenizer differences: OLMo improves significantly on the Qwen reference in both domains ($p$=1.5 $\times$ 10$^{-3}$ Smartphone, $p$=5.3 $\times$ 10$^{-8}$ Beauty) while Llama dips significantly below it ($p < 10^{-10}$, $p$=0.020).

These results support the architectural separation principle being generator-agnostic at the pipeline level: a stronger or weaker frozen generator changes the surface-metric profile, but the attribute model is never re-trained.

## L   Coupling Discovery: Full Rankings

The Phase 1 coupling discovery ranks all $\binom{7}{2}$=21 cross-prefix pairs by the maximum absolute Pearson correlation among their constituent unit pairs. Table 12 reports the complete ranking for both domains. Intra-prefix correlations are omitted as they are induced by one-hot encoding (within a prefix, exactly one unit is active per sample, so pairs are tautologically anti-correlated). The Tag_Auto_* prefix consists of automatically-extracted high-frequency review keywords; couplings involving Auto_* tend to dominate the top of the ranking by tautology (e.g., Tag_Auto_refurbished $\times$ Tag_Topic_Reuse), and we therefore base the discriminative discussion in §6 on the highest-ranked *semantic* pairs (Brand, Price, PriceRange, Purchase, Rating, Topic).

The cross-domain comparison underlies the three-tier structure summarized in §6. Brand $\times$ Purchase, the strongest *semantic* coupling on Smartphone (rank 4, $|r|_{\max}$=0.636, driven by Apple repeat-purchase), is

Table 12: Cross-prefix coupling rankings on Smartphone and Beauty. $|r|_{\max}$ is the maximum absolute Pearson correlation across all unit pairs spanning the two prefixes; $|r|_{\mean}$ averages across the same pairs. Sorted by $|r|_{\max}$ within each domain.

| | Smartphone | | | | | Beauty | | | |
|------|------------|----------|-----------------|-------------------|------|------------|-----------|-----------------|-------------------|
| Rank | Prefix $a$ | Prefix $b$ | $|r|_{\max}$ | $|r|_{\mean}$ | Rank | Prefix $a$ | Prefix $b$ | $|r|_{\max}$ | $|r|_{\mean}$ |
| 1 | Auto | Topic | 0.993 | 0.031 | 1 | Auto | Topic | 0.926 | 0.025 |
| 2 | PriceRange | Purchase | 0.908 | 0.226 | 2 | Price | Purchase | 0.494 | 0.031 |
| 3 | Price | Purchase | 0.782 | 0.113 | 3 | PriceRange | Purchase | 0.289 | 0.035 |
| 4 | Brand | Purchase | 0.636 | 0.037 | 4 | Auto | Brand | 0.287 | 0.006 |
| 5 | Auto | Brand | 0.388 | 0.013 | 5 | Rating | Topic | 0.212 | 0.023 |
| 6 | Brand | Price | 0.277 | 0.053 | 6 | Auto | Rating | 0.202 | 0.023 |
| 7 | PriceRange | Price | 0.264 | 0.085 | 7 | Brand | Topic | 0.161 | 0.011 |
| 8 | Purchase | Topic | 0.259 | 0.026 | 8 | Price | Topic | 0.088 | 0.021 |
| 9 | Rating | Topic | 0.254 | 0.041 | 9 | Auto | Price | 0.084 | 0.012 |
| 10 | Brand | Topic | 0.249 | 0.018 | 10 | Brand | Price | 0.076 | 0.016 |
| 11 | Auto | Purchase | 0.239 | 0.018 | 11 | PriceRange | Price | 0.071 | 0.015 |
| 12 | Auto | Rating | 0.182 | 0.033 | 12 | Brand | Rating | 0.040 | 0.005 |
| 13 | Purchase | Rating | 0.137 | 0.033 | 13 | Purchase | Topic | 0.035 | 0.007 |
| 14 | PriceRange | Rating | 0.125 | 0.038 | 14 | Brand | Purchase | 0.027 | 0.002 |
| 15 | Price | Topic | 0.098 | 0.019 | 15 | Auto | Purchase | 0.026 | 0.004 |
| 16 | Auto | Price | 0.096 | 0.016 | 16 | Purchase | Rating | 0.024 | 0.004 |
| 17 | Brand | PriceRange | 0.068 | 0.015 | 17 | Price | Rating | 0.018 | 0.008 |
| 18 | PriceRange | Topic | 0.052 | 0.016 | 18 | Auto | PriceRange | 0.015 | 0.002 |
| 19 | Auto | PriceRange | 0.051 | 0.011 | 19 | Brand | PriceRange | 0.013 | 0.001 |
| 20 | Price | Rating | 0.045 | 0.012 | 20 | PriceRange | Rating | 0.007 | 0.003 |
| 21 | Brand | Rating | 0.039 | 0.009 | 21 | PriceRange | Topic | 0.007 | 0.001 |

essentially absent on Beauty (rank 14, $|r|_{\max}$=0.027). Conversely, Rating × Topic ranks 9th on Smartphone (0.254) and 5th on Beauty (0.212) with comparable magnitudes, supporting its reading as a cross-domain invariant rooted in consumer psychology rather than market structure.

**Energy ranking of natural vs. swap-perturbed configurations**

Table 13 reports the full energy-ranking results summarized in §6.

Table 13: Mean-field energy ranking of natural vs. swap-perturbed configurations under the six single-attribute clamps used in §7. $\Delta \bar{F} = \bar{F}(v_{\text{clamped}}) - \bar{F}(v_{\text{original}})$; "% lower" is the fraction of paired samples for which the natural configuration has lower energy (chance is 50%).

| | Smartphone | | | Beauty | | |
|-------|-----|----------------|-----------|-----|----------------|-----------|
| **Clamp** | $n$ | $\Delta \bar{F}$ | % lower | $n$ | $\Delta \bar{F}$ | % lower |
| Rating 5→1 | 500 | +4.37 | 83.8% | 500 | +4.90 | 96.4% |
| Price (Premium/Luxury) → (Entry/Budget) | 162 | +11.88 | 99.4% | 500 | +5.81 | 94.6% |
| Brand swap | 500 | +3.90 | 89.6% | 112 | +6.18 | 96.4% |

**Per-pair marginal flips and base-rate confounding**

An earlier version of the Phase-2 validation reported per-direction marginal flips: clamp $a$=1 and report the percentage change of $P(b)$ (and vice versa) on discovered specialty pairs, designated control pairs (empirical $r \approx 0$), and the invariant Rating–Service pair. Table 14 reproduces those numbers for completeness. The per-direction percentages are dominated by marginal base rates: clamping toward a rare unit (the flip $a \mid b$=1

column) inflates the percentage change regardless of the empirical coupling, which is why control-pair flips in that column are comparable to—and on Beauty exceed—the specialty flips. The direction-dependence of the invariant pair (clamping Topic(SERVICE)=1 raises $P$(Rating=1) by +0.83–+1.24% while the reverse direction is near zero or slightly negative) reflects the same base-rate asymmetry compounded with the consumer-psychology direction of the coupling. This confound motivates the base-rate-independent interaction contrast of §6 (Eq. 3); the flip table is retained here to document the observed asymmetric pattern rather than to support a near-null claim.

Table 14: DBM per-direction clamp flips on discovered couplings (superseded by the interaction-contrast analysis of Table 3). *Specialty* pairs are domain-specific positive couplings; *control* pairs have empirical $r \approx 0$; *invariant* pairs hold across domains. "Flip $b\,|\,a$=1" is the percentage change in the marginal probability of $b$ when $a$ is clamped to 1. Percentages are confounded by marginal base rates (see text).

| **Domain** | **Pair** | **Type** | $r$ | flip $b|a$=1 (%) | flip $a|b$=1 (%) |
|---|---|---|---|---|---|
| Smartphone | Apple × Topic(DAMAGE) | specialty | +0.249 | −0.093 | +1.693 |
| | Google × Topic(SCREEN) | specialty | +0.117 | +0.187 | +1.162 |
| | Apple × Topic(GAMING) | control | −0.064 | +0.413 | +1.377 |
| | Google × Topic(BATTERY) | control | +0.001 | +0.354 | +0.882 |
| | Rating=1 × Topic(SERVICE) | invariant | +0.254 | −0.039 | +1.237 |
| Beauty | Sally Hansen × Nails | specialty | +0.158 | +0.861 | +0.690 |
| | Sally Hansen × Color | specialty | +0.161 | +1.236 | +0.702 |
| | Bath & Body Works × Fragrance | specialty | +0.135 | +1.337 | +0.668 |
| | Essie × Color | specialty | +0.103 | +1.406 | +0.760 |
| | Sally Hansen × Fragrance | control | −0.034 | +0.152 | +1.295 |
| | Bath & Body Works × Nails | control | −0.014 | −0.217 | +0.956 |
| | Rating=1 × Topic(SERVICE) | invariant | +0.212 | −0.346 | +0.831 |

## M  Empirical and DBM Correlation Structure

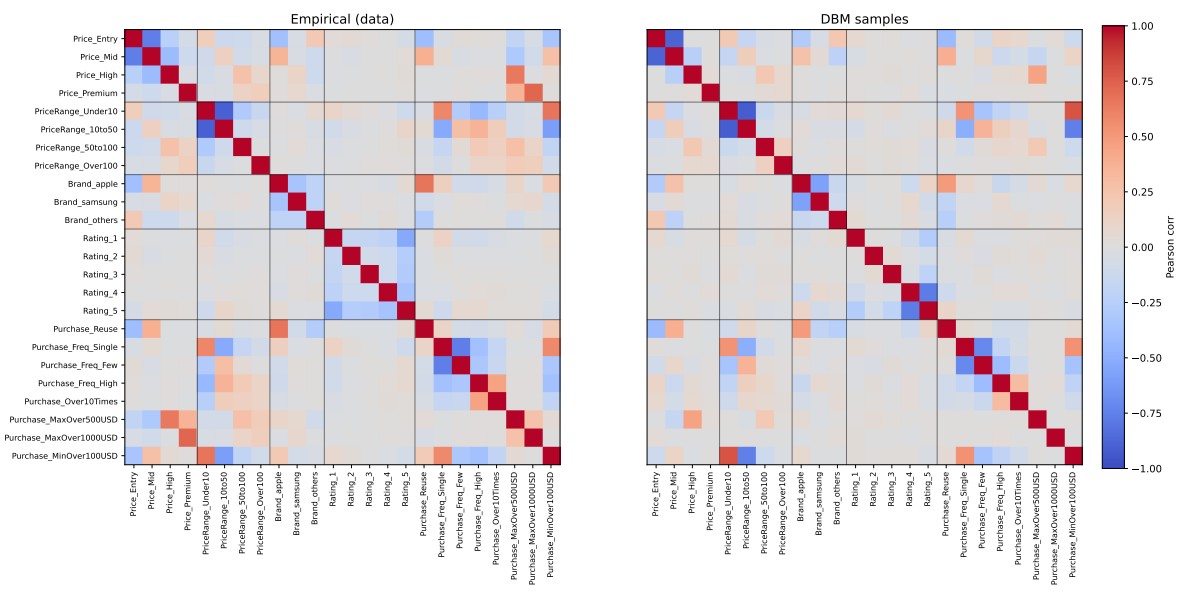

Figure 4: Smartphone: Pearson correlation matrices of visible variables. **Left:** empirical training data ($J$=160). **Right:** samples drawn from the trained DBM. The cross-prefix off-diagonal structure—the brand–price oligopoly block, the rating–service block, the auto-keyword–topic alignment—is preserved by the DBM.

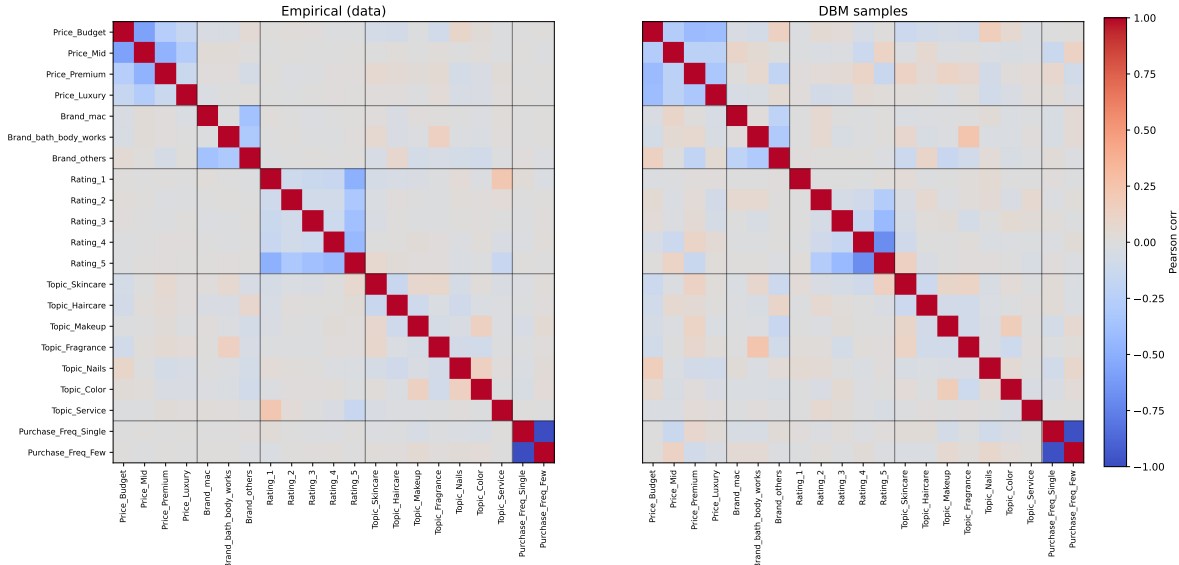

Figure 5: Beauty: Pearson correlation matrices of visible variables. **Left:** empirical training data ($J{=}170$). **Right:** samples drawn from the trained DBM. The off-diagonal pattern is qualitatively distinct from Smartphone—weaker brand–price block, stronger brand–topic specialty block—and the DBM tracks each domain's actual structure rather than imposing a shared prior.

As a complement to the Phase 1/Phase 2 coupling discovery in §6, we report the full Pearson correlation matrices of the visible variables under the empirical training distribution and under samples drawn from the trained DBM. Within-prefix block-diagonal anti-correlations are induced by one-hot encoding and are not informative; the load-bearing structure is the cross-prefix off-diagonal blocks, which encode the market-level co-occurrence patterns the DBM is asked to capture.

The DBM samples reproduce the empirical off-diagonal structure in both domains, including its differences across domains. This is a stronger statement than marginal-matching: it confirms that the DBM has internalised the second-order co-occurrence statistics that the coupling discovery in §6 probes pair by pair.

## N   Qualitative Generation Samples

We illustrate the two clamp axes from §7 with one representative sample each. All texts are verbatim outputs of the proposed model (frozen Qwen2.5-0.5B-Instruct as the generator; no post-editing).

**Sentiment-axis clamp: Smartphone, Rating $5 \to 1$ (Sample 853).** The visible vector $v$ encodes a refurbished smartphone purchase (*Renewed* tag active). Rating is clamped from 5 to 1; all other visible units are held fixed.

**Original**:

> For a refurbished phone, it's not a bad phone. I've bought three phones from this brand and all lasted for months. This phone works great for me. I wish there was a way to get more reviews but they are very positive. I'm not a tech person but I can see myself getting more techs in the future. Thank you. [VIDEO] If you're looking for a refurbished phone, I recommend buying from this brand.

**Clamped**:

> I had a phone factory refurbished and it worked great for 2 months. Then it stopped working. The phone was broken down and sent back to Amazon. The refurbished phone worked fine for 2 months before I returned it. It was like I had a good phone but then I got a bad phone. I was very upset. The Amazon customer service was very bad and took forever to respond. I have to go to another store to get a refund.

**Effect of the clamp.** The model preserves the refurbished framing across both generated texts (shared vocabulary: "refurbished", "brand", "Amazon", repeat-purchase narrative); only the evaluative direction flips. The original is a conditional positive endorsement ("If you're looking for a refurbished phone, I recommend buying from this brand"); the clamped text is a coherent negative narrative (worked for two months, then broke, poor seller response). The clamped generation matches the typical low-rating mode rather than this specific one.

**Content-axis clamp: Beauty, Bath & Body Works $\times$ Fragrance (Sample 397).** The visible vector $v$ encodes a generic beauty product. Both Brand = Bath & Body Works and Topic = Fragrance are clamped together, exploiting the Brand–Topic specialty coupling discovered in §6.

**Original** (target-topic keywords = 0):

> This is a great product. I love it, I love a bottle and I have no problem with the sleeve. My big thanks to Mangiacotti for creating a great product. They are so easy to use. My kids love it. They really take it from a holiday party to a holiday meal. I love this product.

**Clamped** (target-topic keywords = 3):

> Great product with an attached scent. I am really pleased with the quality of the product. I was looking for a gift for a friend that loves all the smells in this scent. I will definitely get this in the future. I love the cutlery and the message of joy. I love it.

**Effect of the clamp.** The positive evaluative framing is preserved: both texts read as enthusiastic recommendations. What shifts is the topical content. The original mentions no fragrance vocabulary (it is a generic enthusiastic review); the clamped text reorganizes around scent ("attached scent", "loves all the smells in this scent") and frames the product as a gift, consistent with the Fragrance category. Target-topic keyword count rises from 0 to 3, sentiment is statistically unchanged.

**Two axes, illustrated.** Together these two samples exhibit the orthogonality claimed in §7. The Rating clamp on Smartphone moves generation along the *sentiment* axis while preserving the contextual framing (refurbished, brand-loyal repeat purchase). The Brand–Topic clamp on Beauty moves generation along the *content* axis while preserving the evaluative framing (positive recommendation). Each clamp exercises one axis without disturbing the other—the qualitative counterpart of the quantitative orthogonality reported in Tables 4, 9, and 5.

