# OpenReview forum: "Energy-Based Attribute Models for Controllable Review Generation with Frozen LLMs"
_TMLR — Under review for TMLR_

### Review · Reviewer_edue · 2026-07-06

**Summary Of Contributions:**

The paper proposes a modular controllable-generation setup: a frozen LLM writes the text, while a separate DBM learns structured review/product attribute relationships and feeds them in through soft prompts. The core idea is useful: give the LLM an inspectable, clampable attribute model instead of relying only on fine-tuning or activation steering. The results are promising for structured review control, especially rating/sentiment control, but the broader "world model" and "meaning" framing is stronger than the evidence supports.

Strengths:
- Clean DBM + adapter + frozen LLM architecture.
- Strong form-vs-control framing with a useful direct-MLP baseline.
- Convincing rating/sentiment control results.

Weaknesses:
- "World model" / "meaning" claims are overstated.
- Content-axis control and control-pair evidence are weaker than claimed.
- Needs stronger structured baselines and less reliance on proxy metrics.

**Audience:**

Yes

**Audience Explanation:**

Despite the minor overclaims, I'd still assume some audience'd generally care about this paper because it tackles a real problem in controllable generation: steering frozen LLMs with an explicit, inspectable structure rather than only fine-tuning, prompt tricks, or activation steering. The DBM + soft-prompt setup is a neat modular idea, and the form-vs-control framing with a direct-MLP baseline should interest people working on controllable text generation, interpretable latent models, PEFT, and energy-based models. Its narrower finding, that explicit joint attribute models can improve structured clamps while preserving generation quality, is still worth knowing.

**Broader Impact Concerns:**

A major concern that I can think of is that this method could make it easier to generate fake or manipulative product reviews with controlled sentiment, brand mentions, prices, or topics. The authors should probably discuss this misuse risk, along with possible safeguards like disclosure, detection/watermarking, privacy considerations around review data, and limits on deceptive synthetic review generation.

**Claims And Evidence:**

No

**Claims Explanation:**

The paper DOES support a narrower claim that _a DBM-style joint attribute model can help structured review control, especially rating/sentiment clamps_, but it does not convincingly support the broader "world model", "meaning" or general controllable-generation claims. Some evidence is messy or overstated: Table 3's control pairs are not clearly near-null, Table 8 seems to contradict the KS claim, and content-axis control improves but remains far from natural target behavior. The baselines are useful but incomplete, since simpler structured models are missing, and the evaluation relies heavily on proxy metrics without human or stronger domain-specific validation.

**Requested Changes:**

1. **Narrow the claims** It would help if the authors framed the method as "explicit joint attribute modeling for controllable review generation," rather than leaning too heavily on "world model", "meaning" or "world understanding."

2. **Add the missing technical details** I’d appreciate it if the authors could include the free-energy definition, prompts, preprocessing, DBM architecture, training details, hyperparameters, seeds, and generation samples, so the main claims are easier to verify.

3. **Fix evidence/prose mismatches** It would help to revisit the "near-null control pairs" claim in Table 3 and correct or qualify the Table 8 KS interpretation.

4. **Add stronger structured baselines** It would make the paper more compelling if the authors compared against simpler explicit joint models, such as RBMs, Ising/log-linear models, Bayesian networks, VAEs, or masked-feature predictors.

5. **Strengthen control validation** Consider adding review-domain sentiment classifiers, attribute classifiers, human checks, and qualitative examples to support the control claims beyond proxy metrics.

---

> ### Author Response · Authors · 2026-07-13
> **Response to Reviewer edue**
>
> We sincerely thank the reviewer for the careful reading and constructive requests. Having re-examined the paper, we agree with the central criticism: several conceptual claims were stronger than the evidence, and two prose–evidence mismatches are real. We will upload the revised PDF with a summary of changes by July 20, within the discussion period. We would be grateful for any further feedback in the meantime.
>
> 1. Narrowing the claims. We agree. In the revision we will (i) retitle the paper to "Energy-Based Attribute Models for Controllable Review Generation with Frozen LLMs", (ii) reframe the contribution as explicit joint attribute modeling for controllable review generation, (iii) remove the ontological form/meaning framing from the abstract and introduction (retaining Bender & Koller only as an organizing analogy), and (iv) restrict "world model" to one explicitly scoped Related Work passage (static attribute co-occurrence only), disclaiming any general world modeling.
>
> 2. Missing technical details. The free-energy definition, prompt template, hyperparameters, and verbatim generation samples are originally in the supplementary material (App. A, B, D, H); we will merge all appendices into the main PDF and document preprocessing, the DBM architecture, and the seed protocol. This also exposed an error: the abstract's "31K-parameter DBM" is a leftover from the earlier (non-archival) workshop version; the DBM used in all reported experiments is [$J_v$, 512, 1024, 2048], ~2.71M parameters. The revision states the full configuration.
>
> 3. Evidence/prose mismatches (Tables 3 and 8). We agree on both counts. The "near-null control pairs" claim is not supported as stated: raw flip percentages are confounded by marginal base rates, and in the a|b=1 direction control flips can exceed specialty flips. We replaced it with a calibrated analysis: a base-rate-independent interaction contrast $d(a,b)=[E(a1,b0)+E(a0,b1)]−[E(a1,b1)+E(a0,b0)]$, related to the empirical coupling r across all 21 cross-prefix pairs. The DBM's energy response tracks the empirical coupling with Pearson $+0.941 \pm 0.013$ (Smartphone) and $+0.802 \pm 0.020$ (Beauty), both significant under a 5,000-draw permutation null ($p=0.0002$). The binary "near-null" claim is dropped; residual control-pair leakage on Smartphone is reported and discussed rather than claimed away. For Table 8 we will correct the prose: the proposed model is not "unrejected by KS anywhere" (Smartphone combined is rejected at $p=0.004$); the accurate statement is best Wasserstein in 3 of 4 cells and the only system with any KS non-rejections, while CAA is rejected in all cells.
>
> 4. Stronger structured baselines. We have implemented two of the suggested baselines under the identical 3-seed protocol; both will be in the revision. (i) RBM: a single-hidden-layer model with total hidden dimensionality matched to the DBM's 3584 units, so the adapter is identical and only depth is ablated. On form (the RBM wins no cell; the DBM wins Smartphone CE and Beauty CosSim significantly), coupling recovery (r-response Pearson +0.91/+0.72 vs +0.94/+0.80), and sentiment-axis clamps including the combined Rating–Service clamp, a single-layer latent model suffices; we will report this plainly. Depth becomes load-bearing in two-axis control: the DBM's content-axis shift is ~1.5x stronger (keyword shift +0.57 vs +0.37; +0.60 vs +0.44), and the RBM breaks two-axis orthogonality. RBM's content clamp leaks significantly into sentiment ($\Delta$RoBERTa −0.048, $p=7.9e-4$) where the DBM remains null. This is a weaker form of the entanglement failure we report for CAA steering, answering "why a DBM": hierarchical latents preserve independent multi-attribute control. (ii) An energy-free correlation-based clamp baseline: the empirical lift $P(b|a=1)−P(b)$ trivially matches the coupling ranking (Pearson +0.92/+0.95 with r), sharpening the point: coupling discovery requires no energy model, but coherence scoring and clamp-and-re-equilibrate generation structurally do (this baseline has neither an energy function nor a generative path). The revision will additionally discuss Ising/log-linear models, Bayesian networks, and VAEs conceptually, which operations each family supports.
>
> 5. Stronger control validation. We are re-scoring all clamp experiments with a review-domain sentiment classifier and topic/attribute classifiers alongside the current RoBERTa scorer; results will be in the revision, and the verbatim qualitative examples (App. H) will be surfaced more prominently. We do not add a human evaluation; we rely on multi-classifier validation and will state this limitation.
>
> 6. Broader impact. We will add a Broader Impact section covering misuse risks of controllable synthetic reviews (deceptive promotion, fake negative reviews, reputation manipulation), plus disclosure, detection/watermarking, and data-privacy safeguards.

---

### Review · Reviewer_FPTi · 2026-07-06

**Summary Of Contributions:**

This paper proposes a modular architecture for controllable text generation that separates “form” and “meaning”: a frozen LLM is used as the surface text generator, while an external energy-based model captures structured latent attribute co-occurrences. Concretely, the method trains a Deep Boltzmann Machine over binary visible features extracted from Amazon reviews, maps its mean-field beliefs into soft prompts using a lightweight MLP adapter, and conditions a frozen Qwen2.5-0.5B-Instruct model for review generation. The central claim is that a direct-MLP baseline should match the proposed model on form-level metrics, while the DBM-based architecture should provide additional meaning-level capabilities such as energy-based coherence scoring, predictively valid clamping, and two-axis attribute control.

# Strengths:

1. The architectural idea is clear and timely: using an explicit, inspectable latent model to condition a frozen LLM is an interesting alternative to monolithic fine-tuning or activation steering.
2. The paper also includes a relatively strong set of baselines, including full fine-tuning, LoRA, CTRL, PPLM, one-shot ICL, direct-MLP, and CAA activation steering.
3. The experiments are extensive for a short submission: the authors evaluate form-level quality, DBM coupling discovery, single-attribute clamps, combined clamps, content-axis clamps, and comparison against activation steering. The direct-MLP baseline is particularly important because it helps separate the benefit of the soft-prompt conditioning channel from the contribution of the DBM itself.

# Weaknesses:

1. The paper’s language around “world models,” “meaning,” and “counterfactuals” is stronger than what the current evidence supports. The DBM appears to learn useful co-occurrence structure in two Amazon review domains, but this is not sufficient to establish semantic understanding, causal counterfactual validity, or general world modeling.
2. The visible features include auto-extracted topical features from the target review, which makes the setup closer to controlled rendering from desired attributes than realistic prediction or generation from product/user context alone. The paper acknowledges this, but the main claims should be better aligned with this setting.
3. The experimental scope is limited to two review domains and a small frozen LLM; the transfer experiments are only briefly mentioned and appear to be single-seed.
4. Some of the “meaning-level” validation results would benefit from stronger statistical treatment, confidence intervals, and clearer ablations.

Overall, I find the paper interesting and potentially suitable for TMLR after revision, but I recommend that the authors temper some of the conceptual claims and strengthen the empirical validation of the DBM’s specific contribution.

**Additional Comments:**

The paper is well written overall and the experimental narrative is unusually coherent: the authors first establish form-level quality, then validate the DBM’s coupling structure, then test controllability and compare to activation steering. This structure makes the paper easy to follow.

My main recommendation is not to discard the current direction, but to sharpen the claim. The work is strongest when presented as an architecture for structured, interpretable, attribute-conditioned review generation with frozen LLMs. It is less convincing as evidence for general LLM world modeling or semantic counterfactual reasoning. With more careful framing, stronger uncertainty estimates for the DBM analyses, and clearer discussion of feature provenance and dual-use risks, this could become a solid TMLR paper.

**Audience:**

Yes

**Audience Explanation:**

The paper should be of interest to researchers working on controllable generation, frozen-LLM adaptation, energy-based models, modular architectures, and interpretability/intervenability. The main idea—conditioning a frozen LLM through an explicit, inspectable energy-based latent model—is a useful direction that differs from common PEFT, prompting, and activation-steering approaches. The comparison with direct-MLP and CAA is also valuable because it highlights an important distinction between surface-level generation quality and structured attribute control.

The audience interest would be stronger if the authors clarified the scope of the claim: this is not yet a general demonstration of world modeling in LLMs, but rather a promising architecture for structured controllable generation in domains where the relevant attributes can be extracted and represented as binary variables.

**Broader Impact Concerns:**

The broader impact section should be expanded. This work directly concerns controllable review generation, which has a clear dual-use risk: it could be used to generate synthetic consumer reviews with controlled sentiment, brand preference, or topic emphasis. Such technology could support legitimate applications such as simulation, data augmentation, or user-preference modeling, but it could also be misused for deceptive product promotion, fake negative reviews, reputation manipulation, or large-scale opinion shaping.

**Claims And Evidence:**

Yes

**Claims Explanation:**

1. The empirical evidence supports the narrower claim that an external DBM over binary attributes can provide useful structured conditioning for frozen-LLM review generation. The form-level experiments are reasonably convincing: the proposed method performs competitively with or better than several adaptation baselines on CE and SBERT similarity, while the direct-MLP baseline behaves largely as predicted. The 3-seed protocol and item-level paired tests are also a strength.

2. The evidence for the DBM’s structured latent modeling is interesting but less conclusive. The coupling-discovery experiment shows that the DBM can recover some domain-specific correlations and respond to clamps, and the energy-ranking experiment suggests that natural configurations receive lower free energy than perturbed ones. However, the reported marginal flips in Table 3 appear small in several cases, and some control pairs still show non-negligible changes. The paper should provide confidence intervals or significance tests for these flips, and should more clearly quantify how much better the DBM is than simpler correlation-based or energy-free alternatives.

3. The strongest concern is conceptual overclaiming. The paper frames the DBM as a “world model” capturing “meaning,” and uses “counterfactual-style” language. Yet the experiments mainly show learned co-occurrence structure in two structured review domains, not causal counterfactual validity or semantic world modeling in a broader sense. The limitations section acknowledges that clamping does not imply real-world causality, but this caveat should be reflected more consistently throughout the abstract, introduction, and conclusion.

**Requested Changes:**

1. Temper the claims around “world model,” “meaning,” and “counterfactuals.”

   The current framing sometimes suggests that the method demonstrates genuine world understanding or causal counterfactual control. The experiments support a more limited but still interesting claim: the DBM learns structured co-occurrence patterns over extracted review attributes and enables controllable generation consistent with those patterns. Please revise the abstract, introduction, and conclusion to avoid implying real-world causal validity or broad semantic understanding.

2. Strengthen the evidence that the DBM contributes beyond soft-prompt conditioning.

   The direct-MLP baseline is useful, but the paper should more directly isolate the DBM’s contribution. For example, compare against an energy-free latent model, a VAE-style latent model, an RBM, a learned embedding table, or a correlation-based clamp baseline. If adding experiments is not feasible, the authors should at least clarify which capabilities are truly unavailable to B1 and which are simply weaker under the reported metrics.

3. Expand or qualify generalization claims.

   The current experiments are limited to two Amazon review domains and a 0.5B frozen LLM. If the authors want to claim a general architecture for controllable text generation, they should include at least one additional non-review domain, one larger LLM, or stronger multi-seed transfer results. Otherwise, the claims should be restricted to structured review-generation settings.

4. Clarify baseline fairness.

   Some baselines receive attributes through text fields, while Proposed and B1 receive a visible vector through an adapter. Please discuss whether this gives different conditioning bandwidths. A soft-prompt baseline with the same visible vector but without DBM, which the paper partly has as B1, should be consistently emphasized as the main fair form-side comparator.

---

> ### Author Response · Authors · 2026-07-13
> **Response to Reviewer FPTi**
>
> We sincerely thank the reviewer for the generous and precise assessment, and for articulating the paper's strongest framing "an architecture for structured, interpretable, attribute-conditioned review generation with frozen LLMs." We would like to adopt it, including in the title. We will upload the revised PDF, together with a summary of changes, by July 20, within the discussion period. All following items will be included; the 3-seed transfer runs (point 3) are already underway.
>
> 1. Tempering "world model," "meaning," "counterfactuals." We agree and will revise throughout. Concretely: the title becomes "Energy-Based Attribute Models for Controllable Review Generation with Frozen LLMs"; "counterfactual(-style)" is replaced by "clamp-based control" everywhere; the causal non-identification caveat currently in §8 is reflected in the abstract, introduction, and conclusion as requested; the form/meaning framing is reduced to an organizing analogy; and "world model" remains only in one explicitly scoped Related Work passage (energy-based lineage; static co-occurrence structure only, with dynamics explicitly out of scope).
>
> 2. Isolating the DBM's contribution beyond soft-prompt conditioning. We have added two baselines: an RBM depth ablation (hidden dimensionality matched to the DBM's 3584 units so the adapter is identical using 3-seed protocol), and an energy-free correlation-based clamp baseline computed from empirical conditionals. The outcome directly answers the reviewer's question of what the DBM contributes beyond the soft-prompt channel, as a three-level capability stratification. (a) Structurally unavailable to B1 and to the correlation baseline: coherence scoring of arbitrary configurations and clamp-and-re-equilibrate generation (the correlation baseline Pearson +0.92/+0.95 with r matches the coupling ranking trivially but has no energy function and no generative path). (b) Available to any latent joint-density model, single-layer or deep: form quality (the RBM ties or narrowly trails the DBM; it wins no cell), qualitative coupling recovery (r-response Pearson +0.91/+0.72 vs +0.94/+0.80), and sentiment-axis clamps including the combined clamp's distributional sharpening. (c) Requiring hierarchical latents: two-axis independent control — the DBM's content-axis shift is ~1.5x the RBM's on both Brand-Topic pairs, and the RBM's content clamp leaks significantly into sentiment on Sally Hansen ($\Delta$RoBERTa −0.048, $p=7.9e-4$) where the DBM stays null. We will state this stratification explicitly in the revision; it is, we believe, a more informative answer than a monolithic "the DBM is better."
>
> 3. Generalization claims. We agree, and will adopt the suggested restriction: all claims will be scoped to structured review-generation settings where attributes are extractable as binary variables (this is also reflected in the new title). In addition, we are upgrading the cross-backbone transfer experiment (App. E; Llama-3.2-1B and OLMo-2-1B) from single-seed to the full 3-seed protocol; the multi-seed numbers will be included in the revision.
>
> 4. Baseline fairness / conditioning bandwidth. We will add an explicit discussion in §4: text-channel baselines receive the same per-sample attributes serialized into the prompt, while Proposed/B1 receive them through the adapter; these channels differ in bandwidth and inductive bias, which is precisely why B1 (i.e., sharing the visible-vector channel) is the leakage-controlled, fair form-side comparator. We will state this role consistently at each comparison site.
>
> 5. Feature provenance. We will move the "controlled rendering of text given desired attributes" characterization (currently §4) into the abstract and introduction so the main claims are aligned with the setting from the start.
>
> 6. Broader impact. We will add a dedicated section following the reviewer's taxonomy: legitimate uses (simulation, augmentation, preference modeling) versus misuse (deceptive promotion, fake negative reviews, reputation manipulation, opinion shaping), plus disclosure, detection/watermarking, and data-privacy safeguards.

---

### Review · Reviewer_YcT7 · 2026-07-08

**Summary Of Contributions:**

This paper presents a controllable review-generation architecture that separates an explicit energy-based world model from a frozen language model. The world model is a Deep Boltzmann Machine (DBM) trained on binary review-derived attributes; a lightweight adapter maps its mean-field beliefs into soft prompts, and a frozen Qwen2.5-0.5B-Instruct model renders the final text. The experiments on Smartphone and Beauty Amazon reviews compare this system with one-shot ICL, a Direct-MLP adapter, full fine-tuning, LoRA, CTRL, PPLM, and CAA activation steering.

The main contribution is a clean form/meaning comparison: the paper predicts Direct-MLP will match or slightly outperform on surface metrics, while the DBM pipeline provides inspectable coupling structure, free-energy coherence scores, and clamp-based meaning controls. The experimental strengths are the leakage-controlled Direct-MLP comparator, multi-seed form-quality evaluation, and meaning-side tests involving coupling discovery, free-energy coherence, single and combined clamps, content-axis clamps, and activation-steering comparison. The main issues are a "near-null" control-pair claim contradicted by the paper's own Table 3 and ambiguous use of natural versus model-consistent reference distributions in a headline clamp result.

**Audience:**

Yes

**Audience Explanation:**

TMLR readers interested in controllable generation, frozen-LLM adaptation, interpretable latent control, energy-based models, and personalized or review generation would learn from this paper.

The useful contributions are the form/meaning comparison with Direct-MLP, DBM free energy as a coherence score, clamp-based tests of sentiment and content handles, and the comparison with CAA steering. The submitted extensions, if real, take it past a bare re-implementation: a newer frozen LLM, a second domain, stronger baselines and multi-seed testing, two-axis clamp analysis, and the CAA comparison. These issues need correction, but they do not remove the paper's reader interest.

**Broader Impact Concerns:**

The paper should include a brief broader-impact discussion of misuse risk from controllable synthetic product reviews. The experiments generate Amazon-style reviews and explicitly control rating, brand, topic, and sentiment/content axes (Sections 4 and 7), which could facilitate deceptive review generation if released without safeguards.

**Claims And Evidence:**

No

**Claims Explanation:**

The architecture and most empirical comparisons are supported, but two claims are overstated or misstated: the near-null control-pair claim and the headline reference-distribution claim for coupled clamps. Consulting Appendices A-H does not change these two concerns: the supplement extends the coupling-discovery and correlation-structure evidence, but does not provide a null calibration for Table 3's control-pair clamp flips or natural-review-text evidence for the Table 6 23% Beauty claim.

Section 3 and Figure 1 specify the DBM, adapter, and frozen-LLM split clearly enough, and the core architectural claim is supported.

On form quality, the prediction holds. Section 1, Table 1, and Section 8 lay out and test the claim that Direct-MLP should match or slightly outperform on form metrics while the DBM pipeline supplies meaning-side tools. Table 1 supports dominance over one-shot ICL, full fine-tuning, LoRA, CTRL, and PPLM. The "ties Direct-MLP" shorthand should be tightened because Direct-MLP wins Smartphone CosSim and Beauty CE by small margins, but the body text already discloses the intended "match or marginally outperform" interpretation. This is a wording issue, not a reason by itself for Q1 = No.

Coupling and coherence are a mixed picture. Tables 2 and 4 back the domain-specific couplings, the cross-domain Rating-Service coupling, and the lower free energy of natural versus swap-perturbed configurations. The "common-null" or "near-null" control-pair claim, though, is contradicted by the paper's own Table 3. The control-pair "flip a|b=1" values are large, not near-null: Apple x Gaming +1.377, Sally Hansen x Fragrance +1.295, and Bath & Body Works x Nails +0.956. For Beauty these control flips exceed every specialty-pair flip in the same column (largest specialty flip a|b=1 is Essie x Color +0.760), so the control pairs are not merely comparable to the discovered pairs but larger. The table also provides no null distribution, statistical test, or threshold defining "near-null." The authors should either provide calibrated evidence for the null-control claim or drop it and report the observed asymmetric pattern instead.

The targeted-clamp results are the other partial case. Sections 7.1-7.4 establish the rating clamps, combined Rating-Service clamps, Beauty brand-topic clamps, and the CAA comparison as evidence for model-internal control relative to Direct-MLP and activation steering. What overreaches is the claim that coupling-aware clamping makes Beauty generations 23% closer to "natural low-rating distributions." Table 6's 23% result uses a model-consistent reference based on naturally conditioned generated samples. Table 8's natural review-text reference does not show the same Beauty reduction from rating-only to combined clamps. The claim should be narrowed to the model-consistent generated reference or backed with natural-review-text evidence.

**Requested Changes:**

- **[Critical]** Substantiate or drop the "near-null controls" / "common-null" claim that Table 3 contradicts (detailed above). Either provide a null calibration, statistical test, or explicit threshold showing the control flips are near-null, or remove the claim and report the observed asymmetric pattern instead.
- **[Critical]** Narrow or substantiate the "23% closer to natural low-rating distributions" claim. Either revise it to say the 23% Beauty improvement is against model-consistent naturally conditioned generated references, as in Table 6, or add evidence that the same improvement holds against natural review-text references.
- **[Strengthen]** Tighten the form-quality wording so it matches Table 1. The paper can claim dominance over one-shot ICL, full fine-tuning, LoRA, CTRL, and PPLM, but should replace unqualified "ties Direct-MLP" shorthand with "matches or is marginally outperformed by Direct-MLP on form metrics" and state that Direct-MLP wins Smartphone CosSim and Beauty CE by small margins.

---

> ### Author Response · Authors · 2026-07-13
> **Response to Reviewer YcT7**
>
> We sincerely thank the reviewer for a precise review including checking the appendices. Both [Critical] findings are correct, and we verified them against our own numbers before responding. We will upload the revised PDF, together with a summary of changes, by July 20, within the discussion period. We hope this gives the reviewer an opportunity to check both Critical corrections against the revised text, and we would be grateful for any further feedback in the meantime.
>
> 1. [Critical] "Near-null controls" (Table 3). We agree the claim is contradicted by our own table, and the reviewer's diagnosis is exact: the raw flip percentages are dominated by marginal base rates (most visibly in the $a|b=1$ direction, where clamping a rare unit inflates flips for control pairs above specialty pairs). We choose the "calibrate" option: in the revision, Table 3 is replaced by an analysis over all 21 cross-prefix pairs relating empirical coupling strength r to the DBM's energy response, using a base-rate-independent signed interaction contrast $d(a,b) = [E(a1,b0)+E(a0,b1)] − [E(a1,b1)+E(a0,b0)]$ precisely because the raw flip percentages are dominated by marginal base rates, as the reviewer diagnosed. Under this calibrated metric the DBM's response tracks empirical coupling with Pearson $+0.941 \pm 0.013$ (Smartphone) and $+0.802 \pm 0.020$ (Beauty), both significant against a 5,000-draw permutation null ($p=0.0002$), with 3-seed dispersion. Two honest caveats will be reported alongside: (i) on Smartphone, control-pair separation remains weak for the DBM (one control pair responds more strongly than some specialty pairs). The leakage identified by the reviewer survives calibration and is discussed; (ii) the asymmetric $a|b=1$ pattern is explained by base rates and reported, not claimed away.
>
> 2. [Critical] The "23% closer to natural low-rating distributions" claim (Table 6). Correct, and we will fix it exactly as suggested. The −23% Wasserstein reduction on Beauty is measured against the model-consistent reference (the model's naturally-conditioned generations), as Table 6's caption defines; against the natural review-text reference (Table 8) the rating-only$\to$combined improvement does not reproduce on Beauty (0.124$\to$0.138). In the revision the abstract and §7.2 will state the claim strictly as an improvement against the model-consistent generated reference, and the natural-text-reference behavior will be reported alongside it rather than left implicit. The distribution-level advantage against natural texts will be claimed only via what Table 8 actually shows (best W in 3 of 4 cells; the only system with KS non-rejections; CAA rejected everywhere).
>
> 3. [Strengthen] Form-quality wording. Adopted verbatim: we will replace the "ties Direct-MLP" shorthand with "matches or is marginally outperformed by Direct-MLP on form metrics," and state explicitly that Direct-MLP wins Smartphone CosSim and Beauty CE by small margins (within seed-to-seed std, as Table 1 reports).
>
> 4. Broader impact. We will add a dedicated Broader Impact section on misuse risks of controllable synthetic reviews (deceptive review generation with controlled rating/brand/topic/sentiment), plus disclosure, detection/watermarking, and data-privacy safeguards.
>
> Beyond the requested changes, and in response to concerns shared with the other reviewers, the revision also narrows the paper's conceptual claims (retitled to "Energy-Based Attribute Models for Controllable Review Generation with Frozen LLMs"; "counterfactual" $\to$ "clamp-based control"), corrects a parameter-count error (the DBM is ~2.71M parameters, not 31K), and adds an RBM depth-ablation baseline and an energy-free correlation-based clamp baseline under the identical 3-seed protocol. In brief: a single-layer RBM matches the DBM on form, coupling recovery, and sentiment-axis clamps, but the DBM's content-axis control is ~1.5x stronger and only the DBM preserves two-axis orthogonality (the RBM's content clamp leaks into sentiment at $p=7.9e-4$ where the DBM stays null); the correlation baseline matches the coupling ranking but supports neither coherence scoring nor clamp-based generation. The latent joint-density model is thus the load-bearing component, with hierarchical depth specifically required for independent multi-attribute control.

---

### Author Response · Authors · 2026-07-18
**Summary of Changes in the Revised Manuscript**

We have uploaded a revised PDF implementing everything promised in our individual responses. Changes are grouped by theme; reviewer tags indicate who requested each item.

1. Claims narrowed throughout (Reviewer edue, FPTi; consistent with YcT7). The paper is retitled *"Energy-Based Attribute Models for Controllable Review Generation with Frozen LLMs."* The model is described as an energy-based attribute (co-occurrence) model; "world model" survives only in the Related Work lineage discussion, explicitly scoped. "Counterfactual" is replaced by clamp-based control; the form/meaning framing is reduced to an organizing analogy (Bender & Koller), and capability claims are stated as "structure-level." The causal non-identification caveat appears in the abstract, introduction, and conclusion; the "controlled rendering" characterization is stated in the abstract; all claims are scoped to structured review-generation settings.

2. Evidence–prose mismatches corrected (Reviewer edue; YcT7 Critical 1 & 2). (i) The "near-null control pairs" claim is withdrawn. Table 3 is replaced by a calibrated analysis over all 21 cross-prefix pairs: a base-rate-independent interaction contrast d(a,b) related to the empirical coupling r (DBM Pearson +0.94/+0.80, permutation p = 2×10⁻⁴). Residual leakage is reported rather than claimed away, including a Beauty control-type pair, surfaced by our own re-analysis, whose response exceeds the median coupled-pair response. The original flip table is retained in the appendix with an explanation of the base-rate confound the reviewers identified. (ii) The "23% closer to natural low-rating distributions" claim is stated strictly against the model-consistent reference (the former Table 6, now Table 9/App. H), and the non-reproduction against natural review texts (0.124→0.138 on Beauty) is reported alongside it in §7.2. (iii) The prose for the former Table 8 (now Table 6) matches the table: best Wasserstein in 3 of 4 cells; the only system with any KS non-rejections (its own Smartphone-combined cell is rejected at p = 0.004); CAA rejected everywhere. (iv) "Ties Direct-MLP" is replaced by "matches or is marginally outperformed by direct-MLP on form metrics," with the two cells direct-MLP wins stated. (v) The "31K-parameter DBM" was a workshop-version leftover; the correct figure (2.71M) is stated.

3. New baselines and ablations (Reviewer edue, FPTi). An RBM depth ablation (single hidden layer, hidden dimensionality matched at 3584 units so the adapter is identical; same data, protocol, seeds) is integrated throughout (Tables 1, 3–5, and 9; setup in App. F). A correlation-lift baseline appears in Table 3, and direct-MLP (B1) rows are added to Tables 4 and 9 so it serves as the fair comparator at every site (Reviewer FPTi). The outcome is a three-level capability stratification (§8): an energy model is required for coherence scoring and clamp-based generation; a latent joint-density model of any depth suffices for form, coupling recovery, and sentiment-axis clamps (on the combined clamp the RBM matches or exceeds the DBM, which we report plainly); hierarchical latents are required for content-axis strength (~1.5×) and two-axis orthogonality (the RBM's content clamp leaks into sentiment at p = 7.9×10⁻⁴ where the DBM stays null). A new appendix extends the stratification conceptually to Ising/log-linear models, Bayesian networks, VAEs, and masked-feature predictors (Reviewer edue).

4. Strengthened validation (Reviewer edue, FPTi). All clamp experiments are re-scored with a review-domain star classifier (nlptown) and a zero-shot topic classifier (BART-MNLI) in a new appendix: the rating clamp moves expected stars from ~4.5 to ~2.0 for all systems, the DBM > RBM > B1 content-axis ordering replicates, and CAA's Fragrance overshoot reappears (0.18→0.93). The cross-backbone transfer experiment (Llama-3.2-1B, OLMo-2-1B) is upgraded to the full 3-seed protocol, with the attribute model held fixed across seeds and backbones: the single-seed ordering and values are fully robust (max deviation 0.004; seed-to-seed std ≤ 0.009), with OLMo-2-1B best on every cell. §8 limitations updated accordingly.

5. Presentation and completeness (Reviewer edue, FPTi, YcT7). All appendices are merged into the main PDF, with new documentation of preprocessing, the DBM architecture and seed protocol, and the CAA setup. A conditioning-bandwidth discussion (§4) makes B1's role as the leakage-controlled form-side comparator explicit. A Broader Impact Statement covers legitimate uses vs. misuse, marginal risk over prompting an instruction-tuned LLM, auditability of the explicit attribute channel, and safeguards. The main text remains within the regular-submission length (12 pages); the energy-ranking table, flip table, and CAA configuration moved to appendices.

We believe every requested change is now implemented in the PDF. We would be grateful for any further feedback during the remaining discussion period.